# Epigenetic control of topoisomerase 1 activity presents a cancer vulnerability

Tae-Hee Lee [1], Colina X. Qiao[1,2,3], Vladislav Kuzin[4,10], Yuepeng Shi[1,2,10], Marina Farkas[5], Zhiyan Zhao[1], Vijayalalitha Ramanarayanan[1,2,6], Tongyu Wu[1,2,8], Tianyi Guan[1], Xianzhen Zhou [1,2,9], David Corujo [5], Marcus Buschbeck [5,7], Laura Baranello [4] & Philipp Oberdoerffer [1] ✉

DNA transactions introduce torsional constraints that pose an inherent risk to genome integrity. While topoisomerase 1 (TOP1) activity is essential for DNA supercoil removal, the aberrant stabilization of TOP1:DNA cleavage complexes (TOP1ccs) can result in cytotoxic DNA lesions. What protects genomic hot spots of topological stress from excessive TOP1cc accumulation remains unknown. Here, we identify chromatin context as an essential means to coordinate TOP1cc resolution. Through its ability to bind poly(ADP-ribose) (PAR), the histone variant macroH2A1.1 facilitates TOP1cc repair factor recruitment and lesion turnover, thereby preventing DNA damage in response to transcription-associated topological stress. The alternatively spliced macroH2A1.2 isoform is unable to bind PAR or protect from TOP1ccs. Impaired macroH2A1.1 splicing, a frequent cancer feature, was predictive of increased sensitivity to TOP1 poisons in a pharmaco-genomic screen in breast cancer cells, and macroH2A1.1 inactivation mirrored this effect. We propose macroH2A1 alternative splicing as an epigenetic modulator of TOP1-associated genome maintenance and a potential cancer vulnerability.

Single-stranded DNA (ssDNA) lesions (SSLs) are among the most abundant DNA aberrations and pose a pervasive threat to genome integrity[1,2]. SSLs frequently arise during transcription- or DNA replication-associated topological stress as a byproduct of DNA topoisomerase 1 (TOP1)-mediated DNA supercoil resolution[3]. TOP1 unwinds DNA by nicking one strand of the double-helix, creating a covalent TOP1:DNA cleavage complex (TOP1cc) and concomitant ssDNA break as transient intermediates[4]. The TOP1 catalytic cycle can be aborted in response to reactive metabolites, aberrant repair in the presence of adjacent base damage or non-B DNA structures, as well as via pharmaceutical TOP1 inhibition[4]. Unresolved TOP1ccs have been linked to somatic mutations, chromosomal aberrations and cell death. Consistent with this, genetic defects in TOP1cc repair can cause human genome instability syndromes, and TOP1cc stabilization is the mechanism of action of various anticancer drugs[3,5–7]. Tight regulation of topoisomerase function is thus essential to ensure genome integrity, and its manipulation presents a potential cancer vulnerability.

Given their complex nature, the resolution of TOP1ccs requires specialized TOP1:DNA adduct removal before the resulting ssDNA gap can be repaired via canonical single-strand break repair (SSBR)[1,8]. Poly(ADP-ribose) (PAR) Polymerase 1 (PARP1) is a multifaceted effector of TOP1cc repair that facilitates DNA:protein adduct degradation,

[1]Department of Radiation Oncology and Molecular Radiation Sciences, Johns Hopkins University School of Medicine, Baltimore, MD, USA. [2]Department of Biochemistry and Molecular Biology, Johns Hopkins Bloomberg School of Public Health, Baltimore, MD, USA. [3]Department of Biology, Johns Hopkins University, Baltimore, MD, USA. [4]Department of Cell and Molecular Biology, Karolinska Institutet, Stockholm, Sweden. [5]Program of Myeloid Neoplasms, Program of Applied Epigenetics, Josep Carreras Leukaemia Research Institute (IJC), Badalona, Barcelona, Spain. [6]RNA Therapeutics Institute, UMass Chan Medical School, Worcester, MA, USA. [7]Germans Trias i Pujol Research Institute (IGTP), Badalona, Barcelona, Spain. [8]Present address: Department of Cell Biology, University of Pittsburgh, Pittsburgh, PA, USA. [9]Present address: Department of Biochemistry, St Anne's College, Oxford, UK. [10]These authors contributed equally: Vladislav Kuzin, Yuepeng Shi. ✉e-mail: PO@jhmi.edu

subsequent hydrolysis of the TOP1cc 3′-phosphotyrosyl DNA bond by Tyrosyl-DNA phosphodiesterase 1 (TDP1), ssDNA end processing as well as SSBR of the lesion[8–10]. PARP1 is primarily activated by ssDNA breaks and DNA:protein adducts near ssDNA gaps[9], promoting the PARylation of itself, the TOP1cc and downstream repair effectors, as well as the recruitment of PAR-binding repair proteins such as the SSBR factor XRCC1[1,8,11,12]. How these interdependent, PAR-mediated repair events are coordinated at the site of damage remains poorly understood.

Chromatin composition has emerged as an effective mediator of genome maintenance[13,14]. However, we know surprisingly little about how chromatin affects the repair of SSLs, including TOP1:DNA adducts. Topoisomerase enzymes are often concentrated in genomic regions prone to recurrent topological stress. The latter emphasizes both the need to promptly resolve torsional constraints before they disrupt the underlying genetic processes, and the potential for epigenetic control of TOP1cc repair[3]. MacroH2A1.1, one of two alternatively spliced isoforms of the macro-histone variant macroH2A1, is the only nucleosome component with an inherent ability to bind PAR[15–17]. Consistent with a role in the orchestration of PARP-driven DNA repair processes, we previously discovered that macroH2A1.1, but not the PAR-binding-deficient macroH2A1.2 splice isoform, acts as an effector of microhomology-mediated end joining (MMEJ) through PARP-dependent interaction with MMEJ repair factors[18]. Moreover, macroH2A1.1 was found to accumulate at the transcription start site (TSS) of active genes[19–21], a hotspot of TOP1 binding and PARP1 activity[22,23].

Here we identify macroH2A1.1 as an epigenetic rheostat of TOP1 function that protects from TOP1-induced DNA lesions by coordinating PAR-dependent TOP1cc repair. MacroH2A1.1 alternative splicing is frequently perturbed in cancer[15], and we find that impaired macroH2A1.1 splicing is predictive of increased sensitivity to TOP1 poisons in cancer cell lines, as well as improved survival outcome in ovarian cancer patients treated with TOP1 inhibitors (TOP1i). Two main conclusions arise from this work: (1) chromatin composition, and specifically TOP1-associated macroH2A1.1 domains, control the response to topological stress, and (2) epigenetic changes via the manipulation of macroH2A1.1 splicing can alter TOP1cc repair outcome, presenting a cancer vulnerability. Our findings have direct implications for transcription-associated genome instability and cancer therapy.

## Results

### macroH2A1.1 defines a TOP1-permissive chromatin environment

Prompted by the observation that both TOP1 and macroH2A1.1 are enriched at the TSS of active genes[21,22], we asked whether colocalization with macroH2A1.1 is a general hallmark of chromatin-bound TOP1. To test this, we performed Cleavage Under Targets and Release Using Nuclease (CUT&RUN) followed by next-generation sequencing (NGS) for macroH2A1.1 and TOP1 in MDA-MB-231 breast cancer cells, which express relatively high levels of macroH2A1.1[24]. To ensure isoform-specific detection of macroH2A1.1, we generated two independent macroH2A1.1 knockout (1.1-KO) clones reconstituted with FLAG-tagged macroH2A1.1[25]. Parental MDA-MB-231 cells or cells reconstituted with empty vector served as negative controls for FLAG-macroH2A1.1 CUT&RUN specificity (Fig. 1a). Replicate NGS experiments were performed for each condition (Spearman correlation coefficient r > 0.9), and good concordance was observed between the two knockout clones (r - 0.7, Supplementary Fig. 1a). Consistent with previous reports, macroH2A1.1 peaks were highly correlated with heterochromatin domains marked by K27-trimethylated histone H3 (H3K27m3), validating our FLAG-IP approach (Fig. 1b, c, Supplementary Fig. 1b)[17,19,21,26]. We then compared macroH2A1.1 and TOP1 chromatin profiles, which revealed robust colocalization of the two proteins at most if not all TOP1-enriched genomic regions, including TSS-proximal TOP1 peaks (Fig. 1a, b). Non-random colocalization of macroH2A1.1 and TOP1 was

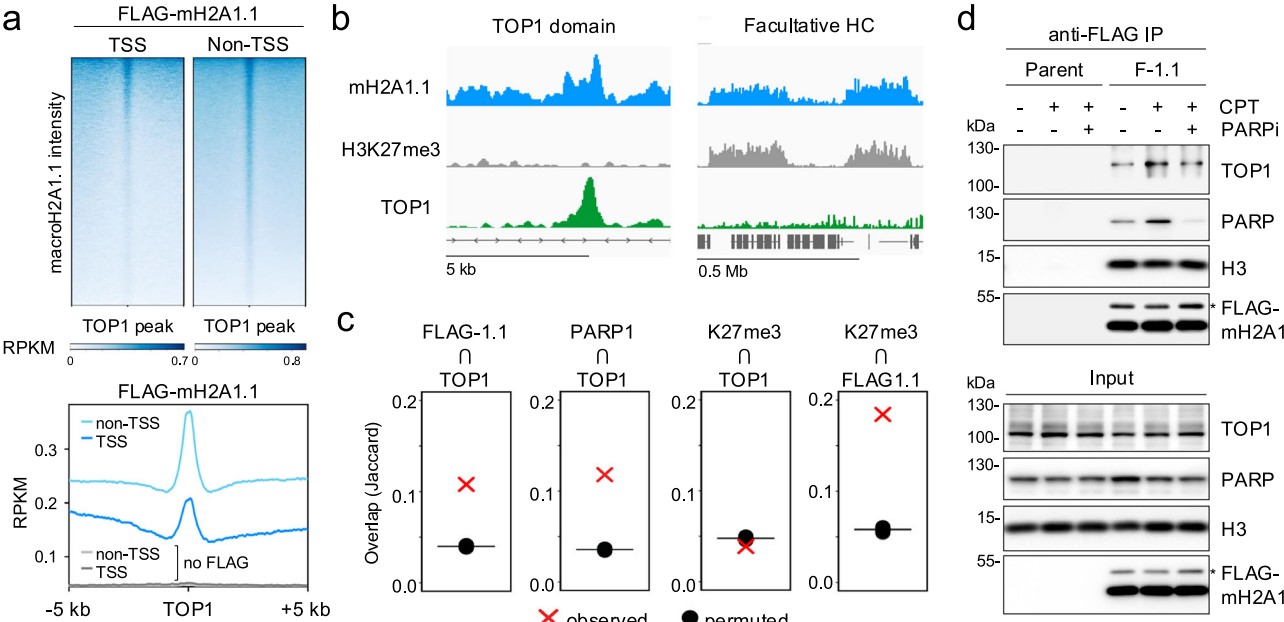

**Fig. 1 | MacroH2A1.1 defines TOP1 permissive chromatin domains. a** Heatmaps and profile plots for FLAG-macroH2A1.1 CUT&RUN signal centered on TSS-proximal (TSS) and TSS-distal (non-TSS) TOP1 peaks in MDA-MB-231 macroH2A1.1 KO cells reconstituted with FLAG-macroH2A1.1. Cells without FLAG-macroH2A1.1 served as a negative control (no FLAG), a representative CUT&RUN experiment is shown. **b** IGV browser shots of distinct macroH2A1.1 chromatin environments associated with TOP1 domains (left) or facultative heterochromatin (HC) marked by H3K27me3 (right). **c** Jaccard indices for observed peak overlap between a feature of interest (top) and a reference feature (bottom), or randomly shuffled reference feature peaks of equal size (permuted). Values on the y-axis represent the intersection divided by the union in base pairs. **d** Western blot for the indicated proteins in nuclear lysates (input) or anti-FLAG IP samples from parental and FLAG-macroH2A1.1 knock-in 293 cells (F-1.1) in the presence or absence of CPT and the PARPi Olaparib. Similar results were obtained in a second, independent experiment. * ubiquitinated macroH2A1. Source data are provided as a Source Data file.

confirmed using the Jaccard index, a statistic for measuring sample set similarity between two peak sets by comparing the overlap between observed and randomly shuffled (permuted) peaks of equal size (Fig. 1c). The overlap of macroH2A1.1 with TOP1 peaks was comparable to that of PARP1, a known interactor of both TOP1 and macroH2A1.1[19,27,28]. TOP1-associated macroH2A1.1 domains were notably distinct from heterochromatin-associated macroH2A1.1 domains, as they were depleted for H3K27me3 (Fig. 1b, c), reminiscent of macroH2A1 regions previously associated with PARP-dependent gene regulation[19,21,29].

To assess if macroH2A1.1 colocalization with TOP1 involves physical association between the two proteins, we performed co-immunoprecipitation (Co-IP) in 293 cells expressing FLAG-tagged macroH2A1.1. We observed robust interaction between TOP1 and macroH2A1.1 upon treatment with the TOP1i camptothecin (CPT), which stabilizes catalytically engaged TOP1ccs. MacroH2A1.1 association with the core histone H3 remained unaltered, demonstrating comparable IP efficiency (Fig. 1d). Little basal or CPT-induced interaction was observed between TOP1 and the alternatively spliced macroH2A1.2 isoform (Supplementary Fig. 1d). Corroborating a role for macroH2A1.1 in the cellular response to topological stress, the CPT-induced increase in TOP1:macroH2A1.1 association was suppressed in the presence of the PARP inhibitor (PARPi) Olaparib (Fig. 1d). Moreover, PARP1 itself showed a similar, PARP activity-dependent increase in macroH2A1.1 association upon CPT treatment (Fig. 1d). Consistent with these findings, interaction of a GST-tagged macroH2A1.1 macrodomain with purified TOP1 protein was dependent on the addition of active PARP1 enzyme in an in vitro pulldown assay (Supplementary Fig. 1c). We therefore conclude that macroH2A1.1 colocalizes with TOP1 in a manner that is enhanced upon TOP1cc-induced PARP1 activation.

## macroH2A1.1 protects from TOP1:DNA lesions

We next sought to determine if macroH2A1.1 affects TOP1cc accumulation and/or resolution. To this end, we performed TOP1 Covalent Adduct Detection sequencing (TOP1 CAD-Seq, Fig. 2a)[30], which maps genomic regions harboring catalytically engaged TOP1. Consistent with our previous findings in other cell lines[22,31], TOP1ccs were depleted from the TSS of active genes relative to the surrounding DNA, despite significant TOP1 accumulation (Fig. 2b), suggesting that TOP1ccs are efficiently cleared to ensure genome integrity at these sites of high torsional stress[32]. To test this possibility, we devised a modified CAD-Seq approach that measures TOP1cc resolution upon damage relative to steady state TOP1cc levels. For steady state TOP1cc detection, cells were briefly treated with 20 µM CPT in the presence of the proteasome inhibitor MG132, which prevents repair-mediated TOP1cc degradation and removal[33]. Negligible CAD-Seq signal was observed in the absence of CPT, demonstrating assay specificity (Supplementary Fig. 2a). To assess TOP1cc levels after prolonged damage, cells were treated with CPT for 30 min in the absence of MG132, causing extended TOP1cc trapping while simultaneously allowing for repair of the lesion. Replicate NGS experiments were performed for each condition and combined for downstream analyses (r ≥ 0.85, Supplementary Fig. 2b). The accumulation of damage-induced TOP1:DNA adducts, ΔTOP1cc, was defined as the ratio of TOP1cc levels after prolonged CPT exposure over steady state levels (Fig. 2c). MacroH2A1.1 depletion resulted in a significant, TSS-proximal increase in ΔTOP1cc compared to control cells, consistent with increased TOP1cc accumulation, or impaired TOP1cc turnover, upon damage (Fig. 2c, d, Supplementary Fig. 2c, d). The extent of the latter directly correlated with gene expression levels and was most pronounced at highly transcribed genes, where torsional stress is maximal (Fig. 2d, top panel)[22,32]. In contrast, even highly expressed genes showed little TOP1cc accumulation (low ΔTOP1cc) in the presence of macroH2A1.1, consistent with effective TOP1cc clearance (Fig. 2d, bottom panel). Extending our findings beyond the TSS,

we observed a similar pattern of macroH2A1.1-dependent TOP1cc turnover across TOP1 peaks genome-wide, which was correlated with the extent of macroH2A1.1 enrichment (Fig. 2e). Of note, macroH2A1.1 loss-associated TOP1cc accumulation extended beyond the peak center at macroH2A1.1[high] TOP1 peaks, consistent with overall broader macroH2A1.1 coverage at these sites (Fig. 2e). No change in TOP1 protein levels was observed upon macroH2A1.1 inactivation (Supplementary Fig. 3a). These findings support a functional link between TOP1cc turnover and macroH2A1.1.

To dissect the underlying TOP1cc repair kinetics, we measured TOP1cc accumulation in total genomic DNA using the Rapid Approach to DNA Adduct Recovery (RADAR)[34], which, like TOP1 CAD-Seq, specifically detects TOP1ccs, but uses DNA dot blot instead of IP/NGS (Fig. 2a). As the high CPT concentration required for efficient adduct detection in TOP1 CAD-Seq resulted in relatively limited TOP1cc lesion turnover (Fig. 2c–e, Supplementary Fig. 2e), RADAR analyses of TOP1cc repair kinetics were instead performed with 1 µM CPT. Cells were treated with CPT for 30 min, followed by drug wash-out and varying recovery times in the presence or absence of either macroH2A1 splice isoform. While macroH2A1 loss did not affect overall TOP1cc induction upon CPT treatment, TOP1cc levels remained elevated 30 min after CPT removal predominantly in macroH2A1.1-depleted cells, indicative of delayed TOP1cc repair (Fig. 2f, Supplementary Fig. 2c, f). Taken together, these findings demonstrate that macroH2A1.1 acts in an isoform-specific manner to regulate TOP1cc clearance at genomic regions of high TOP1 activity.

## PAR-binding-deficient macroH2A1.1 impairs TOP1cc clearance

The selective impact of macroH2A1.1 on TOP1cc repair prompted us to determine if this function is dependent on its ability to bind PAR. To test this, we engineered MDA-MB-231 1.1-KO cells that exclusively express a PAR-binding-deficient macroH2A1.1 mutant (macroH2A1.1 G224E[25,35]) and assessed its impact on TOP1cc turnover following CPT treatment, compared to 1.1-KO cells or cells reconstituted with wild-type macroH2A1.1 (Fig. 3a). While both macroH2A1.1 chromatin enrichment and TOP1 expression were comparable in cells expressing either wild-type or G224E mutant protein (Supplementary Fig. 1a, Supplementary Fig. 3a, b), the PAR-binding mutant displayed a significant delay in TOP1cc repair relative to wild-type macroH2A1.1, as did 1.1-KO cells (Fig. 3a, b). A similar result was observed upon overexpression of wild-type or G224E mutant macroH2A1.1-GFP fusion proteins in MCF7 breast cancer cells, which express inherently low levels of macroH2A1.1 (Supplementary Fig. 3c and Fig. 7a)[24]. Together, these findings support a PAR-dependent role for macroH2A1.1 in protecting from aberrant TOP1cc accumulation.

## macroH2A1.1 links PARP1 activity to TOP1cc repair factor assembly

We next sought to determine how macroH2A1.1 modulates TOP1cc repair. We have previously characterized macroH2A1 isoform interactomes using differential metabolic labeling of 293 cells expressing FLAG-tagged macroH2A1.1 or macroH2A1.2 protein[18]. Supporting a splice isoform-specific role for macroH2A1.1 in response to single-stranded DNA lesions, three out of five macroH2A1.1 interactors, PARP1, LIG3 and XRCC1, represent repair factors that are involved in SSBR[1,8,18]. Given that efficient recruitment of XRCC1 to sites of damage is dependent on PARP1 activity[12,36], we decided to further interrogate the functional interplay between macroH2A1.1 and XRCC1. Using Co-IP, we confirmed that XRCC1 and macroH2A1.1 interact in a PARP-dependent manner, while little interaction was observed with the PAR binding-deficient macroH2A1.2 isoform (Fig. 4a). To determine functional relevance for TOP1cc repair, we assessed XRCC1 recruitment to CPT-induced DNA lesions in the presence or absence of macroH2A1.1, using immunofluorescence (IF) imaging in MCF7 cells. CPT treatment caused a significant increase in nuclear XRCC1 foci, as reported

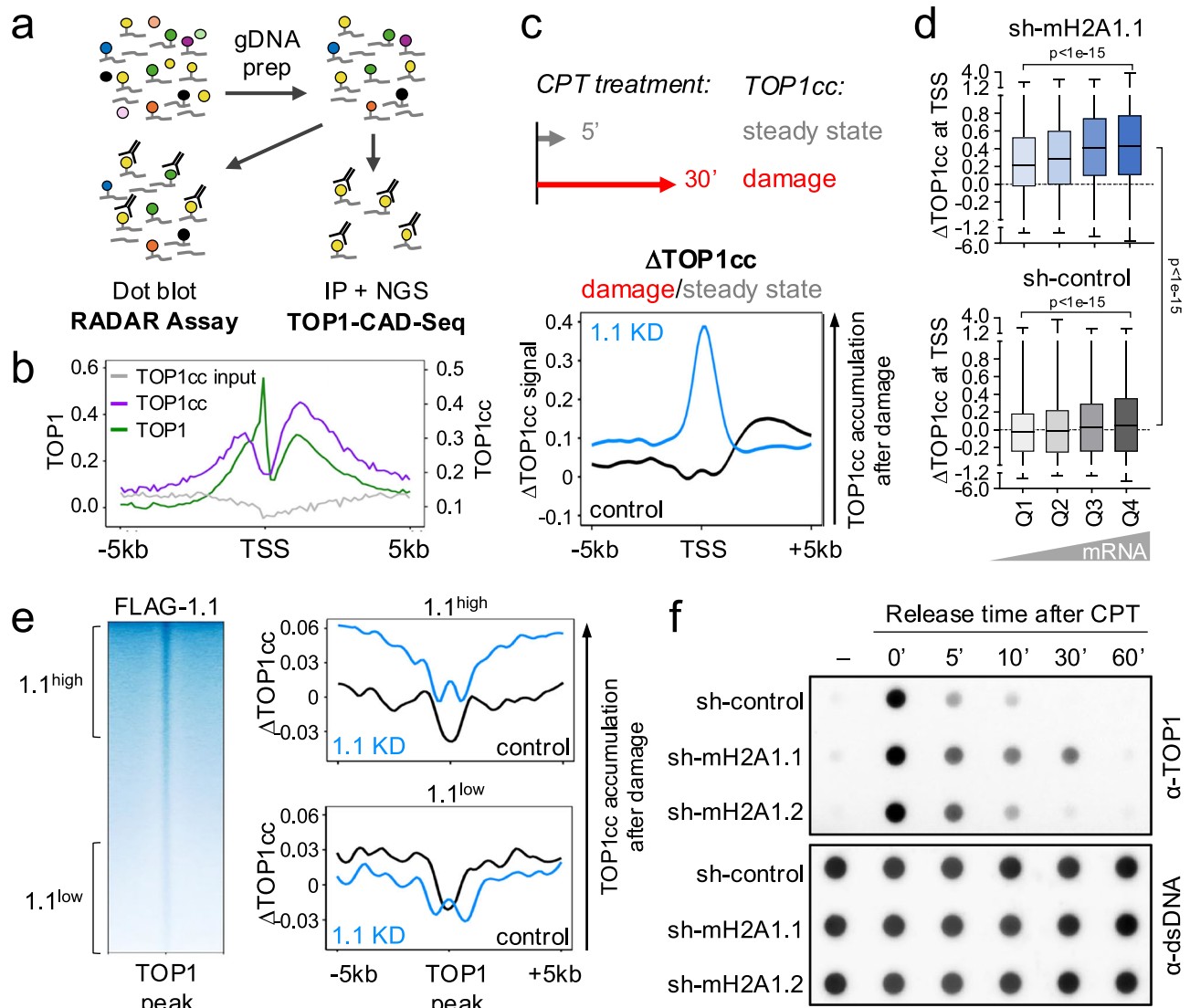

**Fig. 2 | MacroH2A1.1 protects from TOP1cc accumulation. a** Schematic for RADAR and TOP1 CAD-Seq assays. **b** Representative TSS profile plots for TOP1 CUT&RUN (green) and TOP1cc CAD-Seq (purple: IP, gray: input) in MDA-MB-231 cells. Y axes depict Z-normalized read counts. **c** TSS-associated TOP1cc turnover following CPT-induced damage in the presence or absence of macroH2A1.1. Schematic depicts CPT treatment times used to generate the damage. TOP1cc turnover (ΔTOP1cc) is defined as the ratio of prolonged damage (30′ CPT) over steady state (5′ CPT). Profile plot depicts Z-normalized ΔTOP1cc $\log_2$ ratios for sh-RFP control (black) and macroH2A1.1 knockdown (1.1 KD, blue). A ΔTOP1cc $\log_2$ ratio >0 reflects an accumulation of TOP1cc in response to damage. Two independent 30′ CPT and 5′ CPT TOP1 CAD-Seq replicates were combined for average ΔTOP1cc ratios, see Supplementary Fig. 2d for 30′ CPT and 5′ CPT TOP1 CAD-Seq profiles. **d** Comparison of TSS-proximal ΔTOP1cc ratios from (**c**), separated by RNA-Seq-derived gene

expression quartiles; Q1: bottom 25%, Q4: top 25%. Boxplots show the distribution of mean ΔTOP1cc signal (TSS ± 250 bp) within each expression quartile, each data point represents the TSS of one gene. *P* values are based on two-sided Mann−Whitney *U* test for the indicated, pairwise comparisons between TSS quartiles. Box limits represent upper and lower quartiles, whiskers minimum to maximum and center lines represent the median. Similar results were obtained when analyzing replicate experiments independently. **e** TOP1 peak-associated ΔTOP1cc ratios as in (**c**). TOP1 peaks were separated into top (1.1$^{high}$) and bottom tertiles (1.1$^{low}$) based on macroH2A1.1 (FLAG-1.1) enrichment, see TOP1-centered FLAG-1.1 heatmap. **f** TOP1 RADAR assay with genomic DNA from MDA-MB-231 cells expressing the indicated shRNAs, prior to and at the indicated timepoints after CPT treatment (1 μM, 30 min). See Supplementary Fig. 2f for a quantification of two independent experiments. Source data are provided as a Source Data file.

previously[37], which was blunted in macroH2A1.1-deficient cells and restored by wild-type macroH2A1.1 but not the G224E mutant (Fig. 4b–d, Supplementary Fig. 4a, b). Cell cycle profiles and overall XRCC1 expression were comparable across cell lines and experimental conditions (Supplementary Fig. 4b, c). Moreover, defects in XRCC1 foci formation were apparent in non-S phase cells, demonstrating that this observation does not depend on replication-associated topological stress (Supplementary Fig. 4d, e)[38]. Consistent with these findings, we observed a CPT-induced increase in Co-IP of XRCC1 with FLAG-macroH2A1.1, which was dependent on both the macroH2A1.1 PAR binding domain and PARP1 activity (Supplementary Fig. 4f).

As XRCC1 is primarily associated with SSBR downstream of TOP1cc debulking[12], we next asked how these findings relate to impaired TOP1cc turnover observed upon macroH2A1.1 loss (Fig. 2f). The recruitment of the TOP1cc hydrolase TDP1 to TOP1 lesions as well as subsequent TOP1cc hydrolysis are PARP1-dependent[9–11], and both PARP1 and TDP1 are required for efficient XRCC1 recruitment, which in turn further enhances TDP1 activity[36,39]. We thus sought to investigate if macroH2A1.1 loss can alter TDP1 recruitment to damaged DNA. Using chromatin fractionation, we observed a PARP activity-dependent increase in TDP1 association with chromatin following CPT treatment (Fig. 4e, compare lanes 1, 5, 9). In the absence of macroH2A1.1,

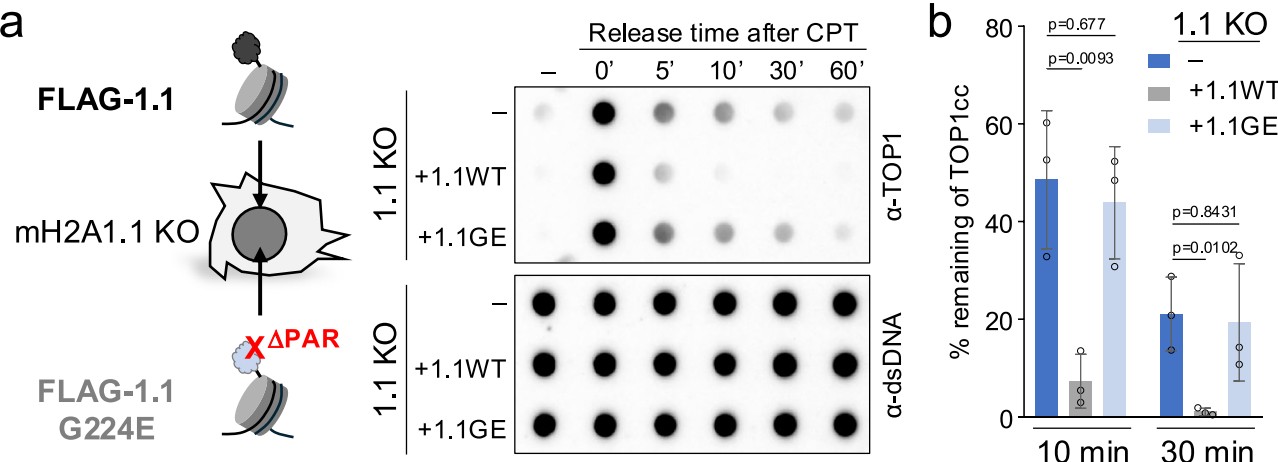

**Fig. 3 | TOP1cc clearance depends on the macroH2A1.1 PAR binding domain.**
**a** TOP1 RADAR assay with genomic DNA from macroH2A1.1 knockout (1.1KO) MDA-MB-231 cells (−) and 1.1KO cells reconstituted with WT (+1.1WT) or G224E mutant FLAG-macroH2A1.1 (+1.1GE), prior to and at the indicated timepoints after CPT treatment as in Fig. 2f, a representative of three independent experiments is shown.
**b** Quantification of RADAR analysis in (**a**), depicting the percentage of remaining TOP1cc relative to 0′ after CPT treatment ($n = 3$ independent replicates), values are expressed as mean and SD. P values are based on two-sided Student's t-test. Source data are provided as a Source Data file.

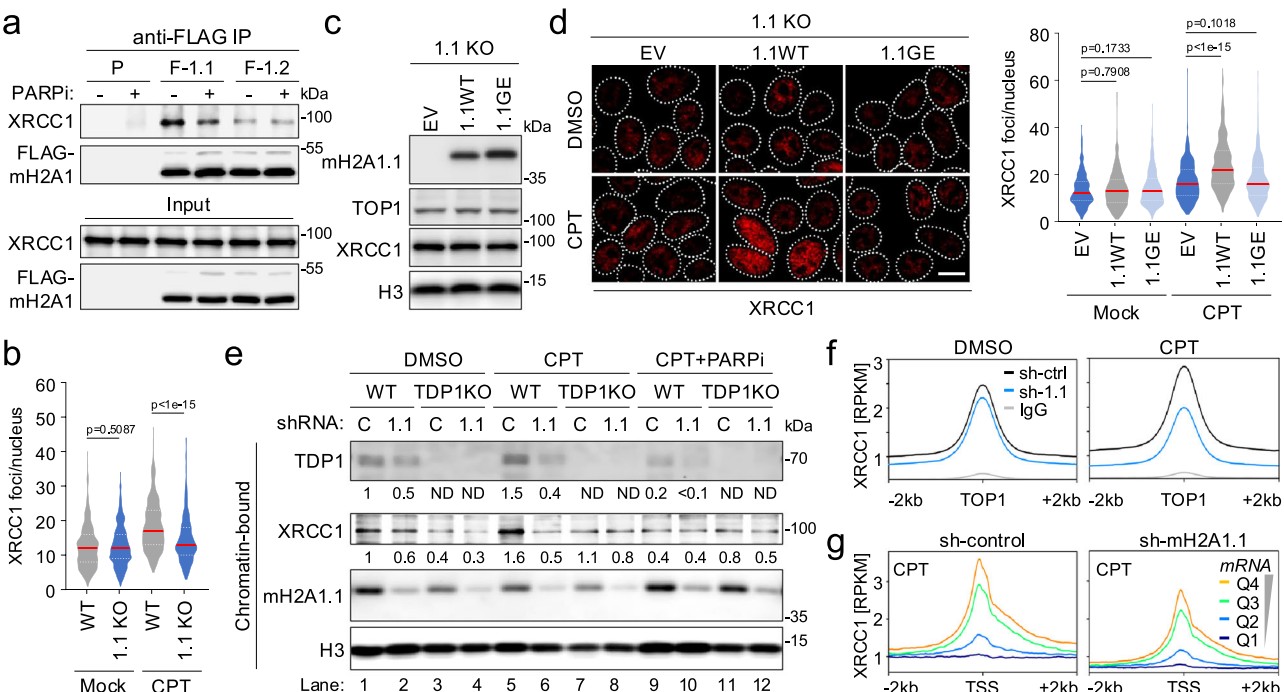

**Fig. 4 | XRCC1 recruitment to TOP1ccs depends on macroH2A1.1. a** Western blot for the indicated proteins in nuclear lysates (input) or IP lysates from parental (P) and FLAG-macroH2A1.1 (F-1.1) or FLAG-macroH2A1.2 (F-1.2) knock-in 293 cells in the presence or absence of PARPi. A representative of two independent experiments is shown. **b** Quantification of XRCC1 foci in MCF7 WT and macroH2A1.1 KO cells in the presence or absence of CPT treatment (1 μM, 30 min), y-axis depicts foci per nucleus (n > 350, see source data for exact n). Similar results were obtained in a second, independent experiment, see Supplementary Fig. 4a for representative images. **c** Western blot for the indicated proteins in macroH2A1.1 KO (1.1 KO) MCF7 cells reconstituted with empty vector (EV), FLAG-macroH2A1.1 (1.1WT) or FLAG-macroH2A1.1 G224E (1.1GE). One of two independent experiments is shown. **d** XRCC1 IF in cells from (**c**) treated with CPT (1 μM, 30 min) or DMSO, scale bar: 10 μm. Foci were quantified as in (**b**), n > 300 nuclei per sample, see source data for exact n. A representative of two independent 1.1 KO reconstitutions is shown. For

violin plots in (**b**) and (**d**), center lines (red) reflect the median, white dotted lines depict upper and lower quartiles, and p values are based on two-sided Mann-Whitney U test for the indicated, pairwise comparisons. **e** Western blot for the indicated proteins in chromatin-bound fractions from WT or TDP1 KO HCT116 cells expressing a control shRNA (C) or sh-macroH2A1.1 (1.1). Cells were treated with DMSO or 1 μM CPT for 30 min, in the presence or absence of PARPi. See Supplementary Fig. 4g for whole cell extracts. Similar results were obtained in a second, independent experiment. Source data for (**a**–**e**) are provided as a Source Data file. **f** CUT&RUN NGS profile plots for XRCC1 or IgG in MDA-MB-231 cells expressing sh-RFP (sh-ctrl) or sh-macroH2A1.1 (sh-1.1), treated with DMSO or 1 μM CPT for 30 min. Averaged RPKM profiles from two independent experiments are shown, centered on TOP1 peaks. **g** Profile plots of CPT-treated cells from (**f**), centered on the TSS and separated based on RNA-Seq-derived gene expression as in Fig. 2d.

chromatin-bound TDP1 was overall reduced and did not increase in response to CPT (Fig. 4e, compare lanes 2, 6, 10). A similar effect was observed for XRCC1 enrichment on chromatin, consistent with our IF data. Moreover, TDP1 knockout did not further impair CPT-induced XRCC1 recruitment in macroH2A1.1-depleted cells, suggesting that TDP1 and macroH2A1.1 are epistatic effectors of XRCC1 recruitment (Fig. 4e, lanes 5–8)[40]. Of note, we were unable to detect a splice isoform-specific interaction between macroH2A1.1 and TDP1 in Co-IP analyses in 293 cells (Supplementary Fig. 4h), suggesting that the effect of macroH2A1.1 on TDP1 is an indirect consequence of XRCC1 and/or PARP1 association with macroH2A1.1. Together, these findings demonstrate that macroH2A1.1 facilitates the recruitment of TOP1cc repair factors in a manner that is at least in part dependent on its ability to bind PAR.

## macroH2A1.1 promotes repair at sites of topological stress

To determine if macroH2A1.1 loss specifically affects repair factor recruitment at sites of endogenous TOP1 activity, we mapped the genomic distribution of XRCC1 with or without CPT treatment using CUT&RUN NGS in the presence or absence of macroH2A1 depletion in MDA-MB-231 cells. Good correlation was observed between replicate NGS experiments as well as control and knockdown cells (Spearman correlation coefficient r > 0.7, Supplementary Fig. 5a, b), suggesting that macroH2A1.1 loss did not alter overall XRCC1 distribution. Analysis of XRCC1 accumulation at TOP1 peaks revealed robust colocalization between the two proteins even in the absence of exogenous DNA damage, pointing to active TOP1cc repair at sites of TOP1 enrichment. In agreement with our chromatin fractionation analyses (Fig. 4e), macroH2A1.1 depletion resulted in a robust reduction of XRCC1 at TOP1 peaks following CPT treatment (Fig. 4f, Supplementary Fig. 5c). A similar effect was observed at the TSS of highly transcribed genes, underlining a role for macroH2A1.1 in the response to transcription-associated torsional stress (Fig. 4g). No XRCC1 signal was detected at non-transcribed TSS, demonstrating assay specificity (Fig. 4g).

To functionally validate our genome-wide data, we sought to more directly monitor the repair of transcription-induced DNA lesions. In contrast to DNA double-strand breaks (DSBs), no tools exist to our knowledge for the imaging-based detection of locally defined TOP1cc repair events. To overcome this limitation, we have adapted a previously described transcriptional reporter system to visualize the accumulation and turnover of TOP1cc repair factors at an actively transcribed genomic locus in U2OS cells[41,42]. Following reporter gene induction with doxycycline (Dox), the site of transcription was detected via the accumulation of yellow fluorescent protein (YFP)-tagged viral MS2 coat protein (YFP-MCP) bound to 24 MS2 stem-loop repeats within the nascent transcript (Fig. 5a). Consistent with previous reports, we observed prominent MS2 foci 5 h after Dox treatment (Fig. 5b)[41,42]. TOP1 Covalent Adduct Detection followed by qPCR revealed robust, Dox-induced TOP1cc accumulation immediately flanking the MS2 TSS, demonstrating efficient TOP1 activation (Fig. 5c). Next, we assessed the accumulation of TOP1 and the XRCC1 repair factor based on average IF signal intensities across at least 50 MS2 sites. An MS2-distal region of equal size was analyzed in parallel to control for background signal intensity (Fig. 5b, d). To prevent PAR chain degradation and thereby stabilize repair factor accumulation upon MS2 induction, cells were treated for 30 min with an inhibitor of poly(ADP-ribose) glycohydrolase (PARGi)[10]. MS2 induction was accompanied by an accumulation of both XRCC1 and TOP1 at and/or adjacent to the MS2 signal, indicative of transcription-associated engagement of the SSBR pathway in response to topological stress. No focal XRCC1 or TOP1 enrichment was observed at the MS2-distal control region (Fig. 5d).

Next, we assessed the impact of macroH2A1.1 loss on XRCC1 accumulation at MS2 foci. SiRNA-mediated macroH2A1.1 depletion caused a significant reduction in XRCC1 intensity at the MS2 peak,

without decreasing overall XRCC1 protein levels or MS2 mRNA induction (Fig. 5e, Supplementary Fig. 6a–c). Consistent with the latter, MS2 peak-associated TOP1 enrichment and TOP1cc adduct formation upon Dox treatment were comparable in macroH2A1.1-depleted and control cells, supporting the notion that macroH2A1.1 loss does not alter transcription-associated TOP1 activity but rather the repair events downstream (Fig. 5c, f). Together, these findings demonstrate that macroH2A1.1 regulates XRCC1 function not only upon genotoxic TOP1 inhibition, but also in response to endogenous DNA transactions.

## macroH2A1.1 protects from TOP1-induced DNA breaks

To probe how macroH2A1.1 loss-associated TOP1cc repair defects affect genome integrity, we assessed DNA break formation following TOP1cc trapping in the presence or absence of macroH2A1.1. Using the alkaline Comet assay, which predominantly detects ssDNA breaks, we observed a significant increase in DNA damage after 1 h of CPT treatment in macroH2A1.1-deficient MCF7 cells, consistent with defective SSBR (Fig. 6a). Underlining PAR dependence, DNA damage following macroH2A1.1 inactivation was suppressed by re-expression of WT macroH2A1.1 but not the G224E mutant (Supplementary Fig. 7a). The effect of macroH2A1.1 loss resembled that observed in TDP1 knockout cells, and no further increase in CPT-induced Comet tail moment was observed in cells deficient for both proteins (Supplementary Fig. 7b), in agreement with the epistatic regulation of XRCC1 recruitment by macroH2A1.1 and TDP1 (Fig. 4e).

Beyond the formation of ssDNA lesions, unresolved TOP1ccs are potent replication and transcription blocks that can produce DNA double-strand breaks (DSBs)[3]. To determine the impact of macroH2A1.1 depletion on CPT-induced DSB formation, we performed IF for S139-phosphorylated histone H2AX (γH2AX). Samples were pulsed with EdU prior to CPT treatment to distinguish cells undergoing DNA replication (EdU+) from non-S phase cells (EdU−). EdU+ cells displayed robust DSB induction within 30 min of CPT treatment, as reported previously[3,43], while few DSBs were detected in EdU− cells under these conditions. However, following release from CPT, EdU− cells showed significant DSB accumulation specifically when macroH2A1.1 was depleted. This observation is consistent with a progressive conversion of unresolved TOP1ccs and/or ssDNA breaks into DSBs, as has been previously observed due to collisions between TOP1ccs and the transcription machinery (Fig. 6b, Supplementary Fig. 7c)[3,44]. Reconstitution of macroH2A1.1 KO cells with WT macroH2A1.1 but not the PAR-binding-deficient G224E mutant efficiently suppressed DSB accumulation in EdU− cells (Fig. 6b). DSB formation in EdU+ S phase cells, on the other hand, was only partially suppressed by macroH2A1.1, pointing to additional, macroH2A1.1-independent pathways that repair TOP1ccs associated with DNA replication (Supplementary Fig. 7d)[3,43]. Together, these findings demonstrate that macroH2A1.1 protects from TOP1cc-driven genome instability.

## macroH2A1 splicing is a marker for TOP1i sensitivity in cancer cells

Defects in TOP1cc repair present a potential cancer vulnerability[45,46]. Exploiting the cytotoxic effects of TOP1cc-associated genome instability, TOP1 inhibition is standard of care for several difficult to treat cancers, including metastatic breast cancer and relapsed ovarian cancer[47,48]. However, the efficacy of CPT and its derivatives varies significantly across cancer types and individuals, and the factors that determine TOP1i responsiveness remain largely unknown. Building on the above observations and the fact that macroH2A1.1 splicing is extensively deregulated in cancer[15,49], we asked whether variability in macroH2A1.1 expression can predict TOP1i responsiveness. Cancer-related macroH2A1.1 isoform variability is recapitulated in NCI60 breast cancer cells, which display up to six-fold changes in macroH2A1.1 protein levels, while expression of the more abundant macroH2A1.2 isoform is largely unchanged (Fig. 7a)[24]. Taking

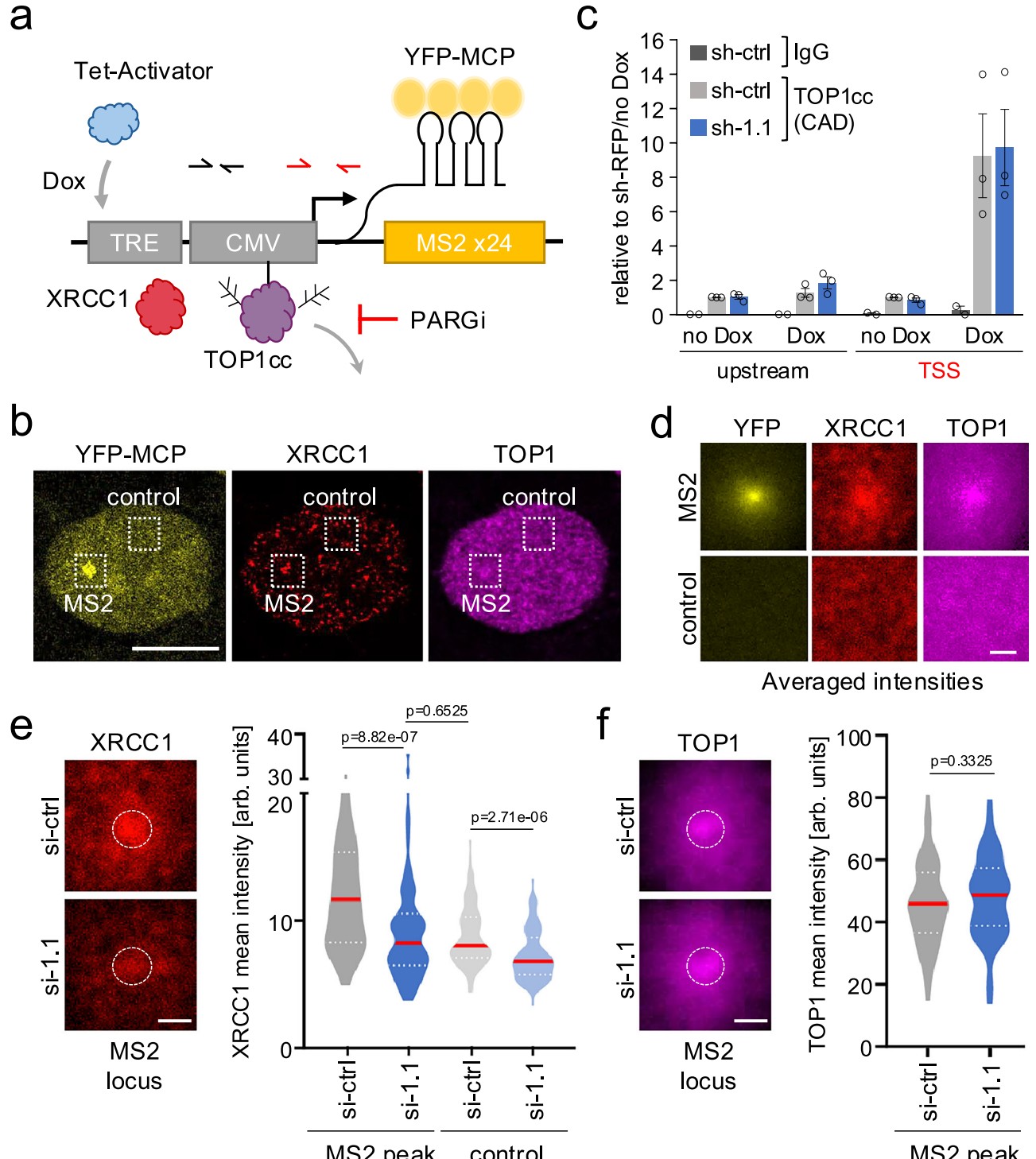

**Fig. 5 | MacroH2A1.1 promotes TOP1cc repair at sites of nascent transcription.**
**a** Schematic of U2OS cell-based reporter for transcription-associated DNA damage repair. An MS2 repeat-containing transcript is induced by Dox and detected with YFP-MCP, PARG inhibitor (PARGi) treatment served to stabilize PAR chains. **b** YFP-MCP, XRCC1 and TOP1 IF images of a representative nucleus 5 h after Dox-treatment, PARGi was added for 30 min prior to analysis. Squares depict the MS2 site or a control region used for analyses in (**d**–**f**), scale bar: 10 μm. Similar results were obtained in three independent experiments. **c** TOP1 CAD-qPCR at the MS2 locus in the presence or absence of Dox in MS2 reporter cells expressing sh-RFP (sh-ctrl) or sh-macroH2A1.1 (sh-1.1). TSS-proximal and upstream qPCR primer pairs are shown in (**a**). IgG was used as a negative control. The y-axis depicts relative enrichment, normalized to TOP1cc levels in untreated sh-RFP cells, average and SEM are from three independent experiments. **d** Averaged signal intensities for the indicated proteins following MS2 induction as in (**a**), n = 82 MS2 or matched control loci.

**e** XRCC1 intensity distributions 5 h after Dox/PARGi treatment in cells transfected with non-targeting control (si-ctrl), or macroH2A1.1 siRNA (si-1.1). For each MS2 site, mean fluorescence intensity was measured at the MS2 peak (white circle) or a corresponding control region as defined in (**b**) (n = 82 MS2 or matched control loci). A representative of two independent experiments is shown in (**d**, **e**); si-ctrl cells were used for analyses shown in (**d**). **f** Average TOP1 intensities and corresponding violin plots as in (**e**) (n > 55 MS2 loci, see source data for exact n). TOP1 signal was measured following 1 h of Dox treatment to avoid potential changes in TOP1 retention due to prolonged repair activity. For all violin plots, center lines (red) represent the median, white dotted lines depict upper and lower quartiles, p values are based on two-sided Mann-Whitney U test for the indicated, pairwise comparisons. Scale bars in (**d**–**f**) are 1 μm. Source data for (**c**, **e**, **f**) are provided as a Source Data file.

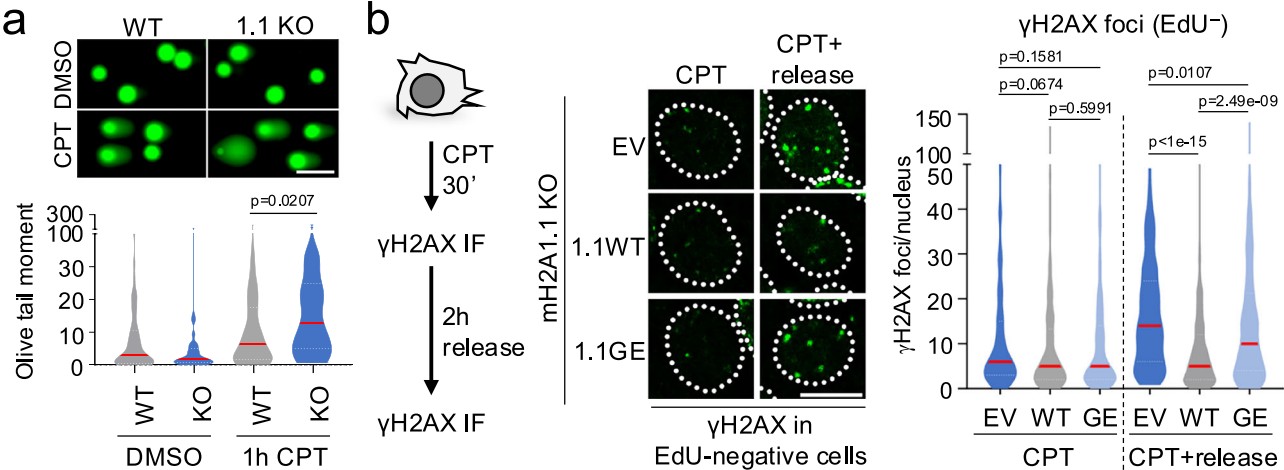

**Fig. 6 | macroH2A1.1 protects from CPT-induced DNA damage. a** Alkaline comet assay in WT and 1.1 KO MCF7 cells treated with 1 μM CPT or DMSO for 1 h; y-axis depicts Olive tail moment (n > 50 nuclei per sample, see source data for exact n), a representative of two independent experiments is shown, scale bar: 100 μm. **b** γH2AX IF in EdU-negative (EdU⁻) macroH2A1.1 KO (1.1 KO) MCF7 cells reconstituted with empty vector (EV), FLAG-macroH2A1.1 (1.1WT) or FLAG-macroH2A1.1 G224E (1.1GE). Cells were treated with CPT (1 μM, 30 min) with or without 2 h

release. Representative EdU⁻ nuclei are shown, scale bar: 10 μm. Violin plots depict γH2AX foci per nucleus (n > 150 nuclei per sample, see source data for exact n). For all violin plots, center lines (red) reflect the median, white dotted lines depict upper and lower quartiles, *p* values are based on two-sided Mann-Whitney *U* test. One of two independent experiments is shown. See Supplementary Fig. 7 for EdU⁺ cells. Source data are provided as a Source Data file.

advantage of a large-scale NCI drug-screening program assessing sensitivity of the NCI60 cancer cell line panel to several thousand compounds including > 200 known DNA-damaging agents[50], we correlated macroH2A1.1 expression with sensitivity to 72 TOP1 inhibitors in the breast cancer subset. Consistent with macroH2A1.1 function as an effector of TOP1cc repair, we observed a robust, inverse correlation between macroH2A1.1 protein levels and TOP1i sensitivity, while TOP1 expression was comparable across cell lines (Fig. 7a, b). Sensitivity to two unrelated groups of chemotherapeutics, tubulin inhibitors and HDAC inhibitors, did not significantly correlate with macroH2A1.1 (Fig. 7b, Supplementary Fig. 8a). Using a colorimetric cell viability assay, we confirmed differential sensitivity to CPT treatment in macroH2A1.1[high] MDA-MB-231 cells and macroH2A1.1[low] MCF7 cells (Fig. 7c). Given that all macroH2A1.1[high] NCI60 breast cancer cell lines were of the triple-negative subtype (TNBC), we expanded our analysis to include MDA-MB-453 TNBC cells, which express ~8 times less macroH2A1.1 than their NCI60 TNBC counterparts (Fig. 7a). MDA-MB-453 cells were significantly more sensitive to CPT treatment than macroH2A.1.1[high] MDA-MB-231 cells, supporting the notion that altered TOP1i sensitivity is not merely a consequence of breast cancer subtype (Fig. 7c). Providing a mechanistic rationale for increased TOP1i resistance in cancer cells with elevated macroH2A1.1 expression, RADAR analysis revealed faster turnover of CPT-induced TOP1cc lesions in MDA-MB-231 cells compared to macroH2A1.1[low] MCF7 cells (Fig. 7d), and TOP1cc clearance in the latter could be improved by macroH2A1.1 overexpression (Supplementary Fig. 3c). Conversely, macroH2A1.1 depletion increased both TOP1cc levels (Fig. 2f) and CPT cytotoxicity in clonogenic and colorimetric survival assays (Fig. 7e, f, Supplementary Fig. 8b). A similar effect was observed in SKOV3 ovarian cancer cells, demonstrating that macroH2A1.1-mediated TOP1cc resistance is not limited to breast cancer cells (Supplementary Fig 8c, d).

Cancer cell sensitivity to TOP1cc trapping agents is thought to result from a combination of TOP1cc accumulation and replication stress triggered by unresolved TOP1:DNA adducts. Both replication stress and TOP1cc repair defects can be exacerbated via inhibition or genetic inactivation of PARP1[45]. We thus asked whether PARPi treatment can overcome relative CPT resistance in TOP1cc repair-proficient macroH2A1.1[high] cancer cells. Co-treatment of MDA-MB-231 cells with the PARPi Olaparib resulted in a dose-dependent increase in CPT-

induced cytotoxicity that was comparable to the cytotoxicity observed in macroH2A1.1[low] MDA-MB-453 cells treated with CPT alone (Fig. 7g). Of note, macroH2A1.1 depletion did not further aggravate the effect of PARPi in the presence of TOP1i, underlining its PARP/PAR-dependent role in TOP1cc repair (Fig. 7f, Supplementary Fig. 8d). Consistent with a macroH2A1.1 isoform-specific TOP1cc repair defect, depletion of the PAR-binding-deficient macroH2A1.2 isoform had only a minor impact on CPT-induced cytotoxicity, which was likely attributable to low levels of replication stress exacerbated by the absence of macroH2A1.2[43] (Fig. 7f). Together, these findings point to high macroH2A1.1 expression as a marker for TOP1i resistance, which can be overcome by simultaneously inactivating PARP.

## MacroH2A1.1 expression has predictive value in cance

To determine potential clinical relevance of our findings, we sought to correlate macroH2A1.1 isoform expression, and by extension TOP1cc repair capacity, with survival outcomes in cancer patients treated with TOP1cc trapping agents. Using the Kaplan−Meier Plotter survival analysis tool[51], we identified a subset of TCGA ovarian cancer patients with macroH2A1.1 splice variant-specific gene expression information that was treated with the TOP1i topotecan[52]. Low macroH2A1.1 expression was significantly correlated with improved survival outcome, consistent with impaired TOP1cc removal in this patient group. Conversely, macroH2A1.1 levels were not indicative of survival outcomes in ovarian cancer patients treated with the microtubule stabilizer taxol (Fig. 7h), in agreement with our cancer cell line data (Fig. 7b), or the nucleoside analog gemcitabine (Supplementary Fig. 8e). These findings suggest that macroH2A1 splicing state may be indicative TOP1i treatment responsiveness.

## Discussion

We have identified macroH2A1.1 as a chromatin effector of TOP1-associated genome maintenance. By coordinating PAR/PARP-dependent repair processes, macroH2A1.1 facilitates the resolution of torsional constraints while simultaneously protecting from aberrant TOP1cc accumulation. This function has clinical implications for therapeutic regimen involving TOP1 inhibition and points to macroH2A1.1 as a predictive marker for cancer cell sensitivity to TOP1 poisons.

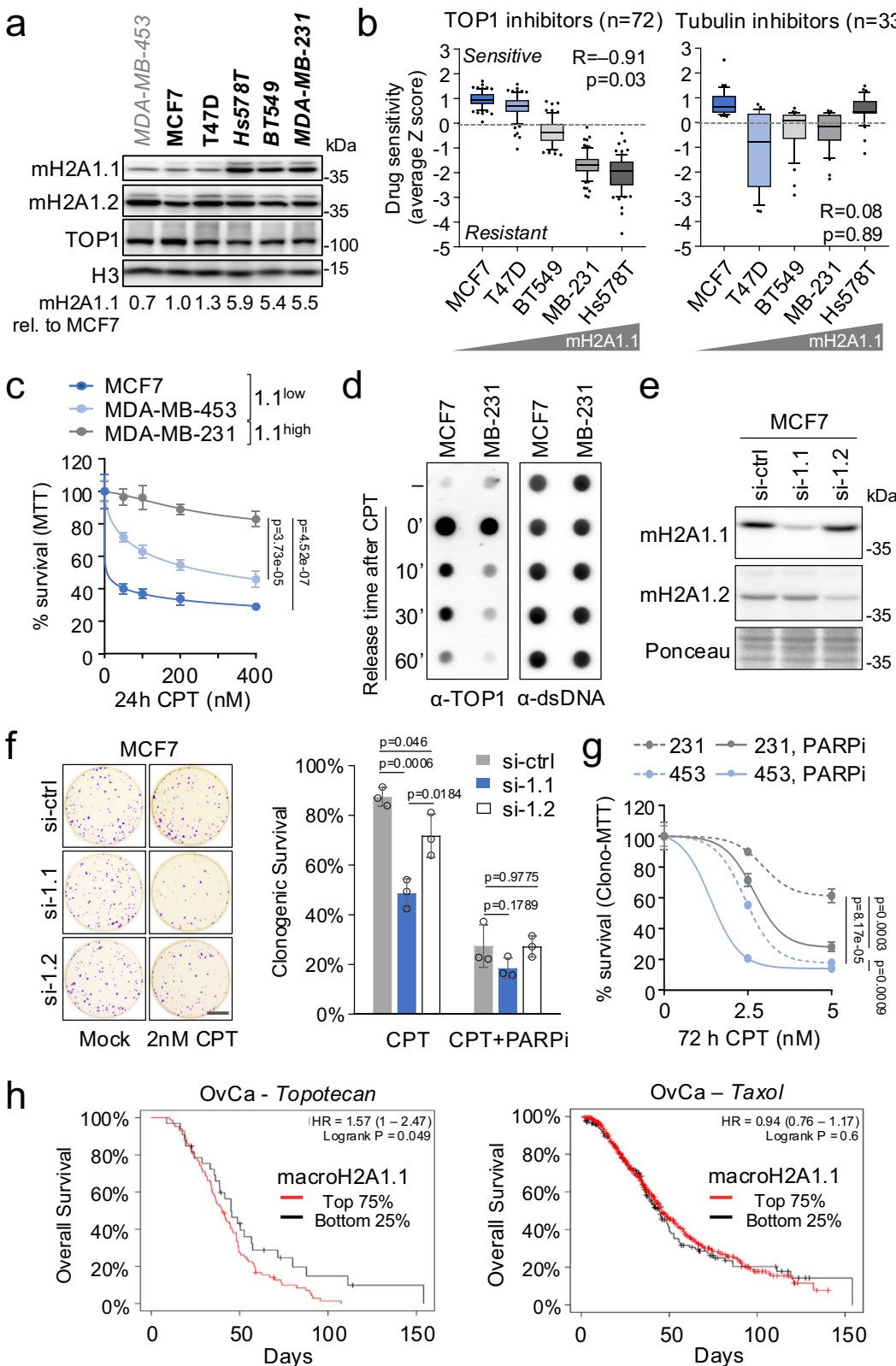

Our findings place macroH2A1.1 at the nexus of the TOP1-PARP1 axis that regulates the cellular response to topological stress. Reminiscent of the enrichment of the homologous recombination (HR)-promoting macroH2A1.2 variant at sites of recurrent replication stress[53], we found that macroH2A1.1 colocalizes with TOP1 domains across the genome. While chromatin remodeling factors have previously been implicated in enhancing TOP1 accessibility to DNA[54],

macroH2A1.1 adds a level of epigenetic control that is directly linked to TOP1 activity, concomitant PARP activation and the recruitment of factors involved in TOP1cc repair. Consistent with this, macroH2A1.1 not only co-occupies TOP1-enriched chromatin domains but interacts with TOP1 in a DNA damage and PARP-dependent manner. By mediating TOP1cc repair at hotspots of TOP1 function, macroH2A1.1 thus provides a rationale for the seeming discrepancy between TOP1

**Fig. 7 | macroH2A1.1 drives TOP1i resistance in cancer cells. a** Western blot for the indicated proteins in MDA-MB-453 (gray) and NCI60 (black) breast cancer cell lines. TNBC cell lines are italicized. A representative of two independent experiments is shown. **b** Drug activity levels in NCI60 breast cancer cell lines based on data from the National Cancer Institute NCI60 drug screening program, arranged by increasing macroH2A1.1 expression, n: number of compounds tested per drug target. Box plots depict average compound activity Z score distributions for each cell line, each data point represents one drug. The majority of compounds were tested in at least two independent experiments, see https://discover.nci.nih.gov/cellminer/ for details. Box limits represent upper and lower quartiles, whiskers the 10 to 90 percentile range, and center lines the median. P values are based on the two-sided Pearson's Correlation Coefficient of median drug activity scores. **c** Cell viability of the indicated cell lines in response to CPT treatment measured by MTT assay, macroH2A1.1[high] cells are in gray, macroH2A1.1[low] cells in blue. Data are presented as mean and SD ($n = 4$ independent replicates); p values are based on two-sided Student's t-test. **d** TOP1 RADAR assay for the indicated cell lines as in Fig. 2f.

Similar results were obtained in three independent experiments. **e** Western blot in MCF7 cells expressing siRNAs against macroH2A1.1 (si-1.1), macroH2A1.2 (si-1.2) or a control sRNA (si-ctrl). A representative of two independent experiments is shown. **f** Clonogenic survival of cells from (**e**) in response to the indicated drug combinations. Survival was normalized to untreated cells for each siRNA transfection. Representative images are shown, scale bar: 1 cm. Data are presented as mean and SD ($n = 3$ independent replicates); p values are based on two-sided Student's t-test. **g** Cell viability in the indicated cell lines in response to CPT treatment in the presence or absence of PARPi, measured by MTT after 10 days of clonogenic growth, data are presented as mean and SD ($n = 3$ independent replicates); p values are based on two-sided Student's t-test. Source data for **a-g** are provided as a Source Data file. **h** Kaplan–Meier survival analysis of TCGA ovarian cancer patient subgroups where treatment regimens contained Topotecan ($n = 119$ patients) or taxol ($n = 821$ patients). Patients were stratified by macroH2A1.1 mRNA expression based on an isoform-specific Affymetrix microarray probe (214500_at), the bottom 25% were considered macroH2A1.1 low expressors.

enrichment and concomitant TOP1cc depletion at sites of recurrent topological stress[22,32]. Although it is conceivable that PARP/PAR helps direct the targeted chromatin incorporation of macroH2A1.1 at TOP1 peaks, we find that macroH2A1.1 accumulation does not require its PAR binding domain (Supplementary Fig. 3b). What controls macroH2A1.1 deposition at sites of TOP1 activity will be an important subject of future investigation. Taken together, our observations provide a rationale for promoter-associated macroH2A1.1 enrichment beyond gene regulation that involves the maintenance of genome integrity. They further suggest that gene expression changes observed upon macroH2A1.1 loss may at least in part be attributed to an impaired resolution of torsional stress[19,21].

Mechanistically, our findings support a role for macroH2A1.1 in the coordination of TOP1cc debulking and subsequent SSBR by providing a platform for the PAR-dependent recruitment and/or retention of the TOP1cc repair factors PARP1, TDP1 and XRCC1. While PARP1 was shown to bind directly to macroH2A1.1 in a PAR-dependent manner[55], other interactions may involve PARylated intermediary factors. For example, auto-PARylated PARP1 may serve as a bridge between macroH2A1.1 nucleosomes and repair factors with conserved PAR binding motifs such as XRCC1[12]. Moreover, XRCC1 was reported to enhance TDP1 activity, which may at least in part contribute to the macroH2A1.1-dependent regulation of TDP1 and TOP1cc debulking[39]. PAR-mediated repair events are likely further enhanced through macroH2A1.1-mediated PAR chain stabilization and concomitant changes in the underlying chromatin structure[25,56]. Together with its recently described roles in MMEJ and oxidative lesion repair[18,56], our findings firmly establish macroH2A1.1 as a chromatin effector of PAR-dependent DNA repair. It will be interesting to determine if macroH2A1.1 has implications for the repair of additional SSBR-dependent, single-stranded lesions.

Using a cell-based reporter for the detection of nascent transcripts, we demonstrate that macroH2A1.1-mediated control of XRCC1 recruitment is relevant in the context of transcription-induced topological stress (Fig. 5c, d). This notion is further supported by genome-wide assays demonstrating a TSS-associated decrease in XRCC1 accumulation and TOP1cc turnover upon macroH2A1.1 depletion (Figs. 2c and 4g). We anticipate that imaging-based interrogation of TOP1-associated repair factors, including but not limited to XRCC1, will significantly advance our understanding of the molecular events that underlie the highly dynamic resolution of torsional constraints.

Given the distinct repair functions of macroH2A1 splice isoforms[18,53,56–58], we propose alternative macroH2A1 splicing as a means to adjust the chromatin environment to changing genome maintenance needs. MacroH2A1.1 expression was found to increase upon cellular differentiation and in postmitotic cells[15,17,59], where the accumulation of SSLs is a main source of deleterious damage that is often associated with degenerative diseases[1,60]. Conversely, expression

of the HR-promoting, replication stress-protective macroH2A1.2 isoform is elevated in replicating cells, including many cancers[15,17]. The two isoforms may furthermore play complementary roles in solving the problem of topological stress, as TOP1cc repair can involve homologous recombination during S phase, owing to the formation of DSBs that are produced by the collision of replication forks with unresolved TOP1ccs[4]. Consistent with this notion, both macroH2A1 isoforms often co-occupy the same chromatin domains[19–21,53], and two chaperones recently identified to promote macroH2A1 chromatin deposition do not distinguish between macroH2A1 splice isoforms[61,62]. It will nevertheless be interesting to investigate if macroH2A1.1-specific control of TOP1cc repair is linked to a distinct and/or unique chromatin remodeling pathway.

Aberrant TOP1cc accumulation is associated with various genome instability syndromes and is exploited as a genotoxic vulnerability in cancer therapy[3,6,7]. Consistent with this notion, we observed increased TOP1 inhibitor-induced DNA break formation in cells depleted for macroH2A1.1 (Fig. 6). Moreover, our findings in cancer cell lines and TCGA patient samples suggest that macroH2A1.1 expression is predictive of TOP1cc repair capacity, which may ultimately help refine TOP1i treatment strategies. For example, macroH2A1.1[high] cancers may benefit from TOP1i/PARPi combination treatment, whereas macroH2A1.1[low] cancers appeared inherently susceptible to TOP1i alone (Fig. 7g). It will further be interesting to determine if manipulation of macroH2A1.1 splicing can help enhance TOP1i sensitivity. Beyond cancer treatment, macroH2A1.1 may protect from repeated TOP1 cleavage, particularly at actively transcribed genes, which were recently shown to accumulate a unique TOP1 mutation signature that contributes to mutagenesis and malignant transformation[6,63].

Altogether, we have uncovered macoH2A1.1 as an epigenetic rheostat of TOP1 activity. Deregulation of this pathway is tied to sensitivity to TOP1 DNA lesions and may be therapeutically exploited as a cancer vulnerability.

## Methods
### Cell lines
Human breast cancer cell lines MCF7 (ATCC), MDA-MB-231 (ATCC), MDA-MB-453, Hs-578T, T47D and BT-549 (gift from D. Gilkes, JHU), as well as HEK293 T-Rex, HEK293 cells (gift from The Broad Institute, Cambridge), HCT116 colorectal cancer cells with or without TDP1 knockout (gift from Y. Pommier, NCI) and U2OS 2-6-3 cells (gift from R. Greenberg, U. Penn) were grown in Dulbecco's modified eagle medium (DMEM, Invitrogen) supplemented with 4.5 g/L D-glucose and 10% FBS (Gemini) and maintained at 37 °C with 5% $CO_2$. All cell lines were regularly tested for mycoplasma using Mycoplasma PCR detection kit (Abcam). For transient knockdown, siRNAs were transfected using DF-1 reagent following the manufacturer's instructions (Dharmacon) and analyzed at 72 - 96 h post transfection. For stable knockdown, lentiviral

infection of LKO.1 shRNA-expression vectors was carried out by spin infection (2500 rpm, 90 min, Eppendorf 5810 R centrifuge) with 8 µg/mL polybrene (Sigma). Cells were incubated overnight prior to virus removal and selection with puromycin (1–2 µg/mL, Invitrogen). For stable knockdown in puromycin-resistant U2OS 2-6-3 cells, selection was performed with high-dose puromycin (30 µg/mL), which efficiently eliminated non-transduced cells. CRISPR/Cas9 knockout of macroH2A1.1 exon 6b was performed using two exon-flanking guide RNAs. In brief, crRNA, tracrRNA and GFP-tagged SpCas9 protein were assembled into an RNP complex, followed by nucleofection (Amaxa Nucleofector) and single cell sorting of GFP$^+$ cells. Clones were screened by PCR for homozygous gene inactivation, cells with intact exon 6b alleles served as clonal wild-type controls. See Supplementary Table 1 for sequence information.

### Antibodies

Primary antibodies (see reporting summary for details): α-macroH2A1.1 (Cell Signaling, Cat#12455: WB), α-macroH2A1.2 (Millipore, Cat#MABE61: WB), α-FLAG (Sigma, Cat#F1804: WB, CUT&RUN, Co-IP), α-TOP1 (BD Biosciences, Cat#556597: WB, RADAR; Abcam, Cat#ab109374: CAD-Seq, CUT&RUN, IF), α-TDP1 (Santa Cruz, Cat# sc-365674: WB; Bethyl, Cat# A301-618A: Co-IP), α-GAPDH (Santa Cruz, Cat#sc-32233: WB), α-H3 (Cell Signaling, Cat#9715: WB), α-XRCC1 (Santa Cruz, Cat#sc-56254: WB, IF; Novus, Cat# NBP1-87154: CUT&RUN, IF), α-γH2AX (Sigma, Cat# 05-636: IF); α-dsDNA (Abcam, Cat#ab27156: RADAR). Secondary Antibodies: goat anti-mouse IgG-Alexa 594/647, goat anti-rabbit IgG-Alexa 647 (Invitrogen: IF); goat anti-mouse IgG-HRP, goat anti-rabbit IgG-HRP (Jackson ImmunoResearch: WB).

### Plasmids

pLKO.1-puro-based shRNA expression vectors were published previously or generated following the provider's instructions (Addgene), see Supplementary Table 1 for shRNA sequences. pLVX-FLAG-macroH2A1.1 vector was published previously[25], pLVX-FLAG-macroH2A1.1 G224E was generated from the latter by replacing a 1176 bp XhoI/XbaI fragment with a G224E (GAG > GGT) mutation-containing gBlock using Gibson cloning.

### Rapid approach to DNA adduct recovery (RADAR)

RADAR was performed essentially as described previously[34]. $1 \times 10^6$ cells were treated with 1 µM CPT for 30 min, and then released into media for the indicated times. Cells were then washed with PBS (GIBCO) and lysed with 600 µL DNAzol (Invitrogen), followed by ethanol precipitation. DNA pellets were solubilized in 8 mM NaOH at 4 °C overnight, then heated at 65 °C for 5 min, followed by shearing with sonication (Fisher Scientific Sonic Dismembrator). DNA content was quantified using NanoDrop One (ThermoFisher), diluted with 25 mM NaPO$_4$ (pH 6.5) to 1–2 µg of DNA per sample, then vacuum-blotted onto nitrocellulose membrane (GE Biosciences) using the Bio-Dot apparatus (Bio-rad). The membrane was washed with 2 × SSC, dried and irradiated with 120 mJ/cm$^2$ using a UV Stratalinker 2400 (Stratagene). The membrane was immunostained using TOP1 and dsDNA antibodies. HRP-conjugated secondary antibodies were used for signal detection by enhanced chemiluminescence (Advansta). Images were captured with the ChemiDoc MP imaging system (Bio-Rad).

### Alkaline comet assay

The alkaline comet assay was performed according to the Trevigen CometAssay™ kit protocol. Cells were treated with 1 µM CPT for 1 h, then trypsinized at 37 °C for 2 min. The cells were centrifuged and resuspended in ice-cold PBS and 500,000 cells/mL were mixed with LMAgarose (R&D systems) at 37 °C at a ratio of 1:10 (v/v). The cell/agarose mixture was transferred onto CometSlides (R&D systems), solidified and immersed in prechilled lysis solution (R&D systems)

overnight at 4 °C. For DNA denaturation, the slides were immersed in alkaline unwinding solution (200 mM NaOH, 1 mM EDTA) for 20 min at RT, followed by electrophoresis in alkaline electrophoresis solution (200 mM NaOH, 1 mM EDTA) at 4 °C. After electrophoresis, the slides were washed twice in dH$_2$O, once in 70% EtOH, dried and incubated with SYBR® Gold in TE buffer for 30 min. The slides were mounted with ProLong Gold Antifade Mountant (ThermoFisher) and imaged using EVOS M7000 Imaging System (Invitrogen). Image analysis was performed using OpenComet in ImageJ.

### Immunofluorescence imaging and analysis

For immunofluorescence (IF), MCF7 or U2OS 2-6-3 cells were plated on poly-L-lysine-coated coverslips 24 h prior to the respective treatments and fixed with 4% formaldehyde for 10 min at RT. Cells were permeabilized with 0.5% Triton-X in PBS for 10 min, washed with PBS-T (0.1% Triton-X) and blocked with 20% FBS in PBS-T for 1 h at RT. After blocking, cells were washed and incubated with primary antibodies for 1 h at 37 °C. Cells were washed and incubated with secondary antibodies diluted in PBS-T, 5% FBS for 1 h at RT. Coverslips were immersed in PBS with 5 µg/mL of Hoechst 33342 (Sigma-Aldrich) for 5 min and mounted on slides with ProLong Gold Antifade Mountant (Thermo-Fisher). For TOP1cc trapping, cells were treated with 1 µM CPT treatment for 30 min, followed by a 2 h release time where indicated. For S phase staining, MCF cells were pulse-labeled with 10 µM EdU for 30 min prior to CPT treatment. After fixation and permeabilization, cells were subjected to Click chemistry labeling (Click-It Kit, Invitrogen) and stained with 5 µM AZDye 647 Azide Plus (Vector Laboratories) for 30 min at RT. Images were captured with Zeiss microscopes Axio Imager Z1 (acquisition software ZEN3.7) or LSM 900 with Airyscan2 (acquisition software ZEN3.6) and analyzed using ImageJ. For XRCC1 and γH2AX foci analysis, 150–500 nuclei were randomly collected and nuclear foci were identified using the 'Find maxima' function in ImageJ. For MS2 intensity analyses, a - 4 × 4 µm square was centered on the MS2 peak or a randomly selected MS2-distal nuclear control region, at least 50 regions were collected from each group following MS2 induction with doxycycline (2 µg/mL, 1 h or 5 h). Cells were treated with 10 µM PARGi PDD 00017273 (MedChemExpress) for 30 min prior to sample collection to prevent PAR chain degradation. "Image sequence" function (ImageJ) was used to combine images from the same color channel into a single TIFF file. To determine TOP1, XRCC1 or MS2 signal intensities at the MS2 peak, mean intensities within a 1 µm-wide circle centered on the MS2 peak were measured for all stacked images. Corresponding nuclear background was defined as the average intensity within the 4 × 4 µm control region of the same cell.

### Cellular extract preparation and immunoblotting

$10^6$ cells were lysed in RIPA lysis buffer (25 mM Tris-HCl, pH 7.5; 150 mM NaCl; 2 mM EDTA; 1% NP-40; 1% Na-deoxycholate; 0.1% SDS) supplemented with cOmplete™ EDTA-free protease inhibitor cocktail (Roche). Lysates were sonicated, centrifuged, diluted with 5 × sample buffer (312.5 mM Tris-HCl, pH 6.8; 10% SDS; 50% glycerol; 12.5% β-mercaptoethanol; 0.05% bromophenol blue) and heated for 10 min at 95 °C. Lysates of equal protein amount based on BSA assay (Bio-Rad) were separated by SDS-PAGE and subjected to western blotting using the indicated primary antibodies. HRP-conjugated secondary antibodies were used for signal detection by enhanced chemiluminescence (Advansta). Images were captured with the ChemiDoc MP imaging system (Bio-Rad).

### Chromatin fractionation

For chromatin fractionation, $0.5–1 \times 10^7$ cells were lysed in ice-cold cytoplasmic buffer (10 mM HEPES-KOH pH 7.6, 3 mM MgCl$_2$, 40 mM KCl, 2 mM DTT, 5% glycerol, 0.5% NP-40, cOmplete™ protease inhibitor cocktail (Roche)) for 10 min on ice, the lysates were centrifuged at

$1250 \times g$ for 5 min at 4 °C. The pellet was washed once with cytoplasmic buffer, centrifuged at $1250 \times g$ for 5 min at 4 °C and resuspended in nuclear lysis buffer (10 mM HEPES-KOH pH 7.9, 0.1 mM EGTA, 1.5 mM MgCl$_2$, 420 mM NaCl, 0.5 mM DTT, 25% glycerol, cOmplete™ protease inhibitor cocktail). Nuclear lysates were incubated on ice for 10 min, and centrifuged at $1250 \times g$ for 5 min at 4 °C. The chromatin fraction-containing pellet was resuspended in N-buffer (20 mM Tris-HCl pH 7.5, 1 mM CaCl$_2$, 100 mM KCl, 0.3 M sucrose, 2 mM MgCl$_2$, 0.1% Triton X-100, cOmplete™ protease inhibitor cocktail) containing a DNA shearing enzyme cocktail (DNase I (10 U/µL, Promega), micrococcal nuclease (40 U/µL, NEB), RNase A (20 ng, Qiagen), and BSA (1:100, Promega). Chromatin was solubilized by pipetting and incubated at 30 °C for 30 min. The lysates were centrifuged at $13{,}000 \times g$ for 10 min at room temperature, and the supernatant was collected as the chromatin fraction. Protein concentrations were determined using the Bradford assay.

## Co-immunoprecipitation

HEK293 T-REx cells expressing Flag-tagged macroH2A1.1 or macroH2A1.2 used for co-IP experiments were described previously[18], parental cells were used as a control. $10^7$ cells were treated as outlined in the figure legends, CPT treatment was at 1 µM for 30 min, PARPi at 10 µM for 24 h. For co-treatment of CPT and PARPi, cells were pre-treated with 10 µM PARPi for 30 min, followed by CPT treatment. Equal numbers of cells were used for each IP experiment. Briefly, cells were washed with PBS and resuspended in hypotonic buffer (20 mM HEPES; 10 mM KCl; 2 mM MgCl$_2$; 10% glucose; 0.5% NP-40) followed by MNase digestion (0.2 U/µL, 4 °C, 60 min) in digestion buffer (20 mM HEPES; 150 mM KCl; 10% Glycerol; 3 mM CaCl$_2$). 20 mM EGTA was used to terminate the reaction. Samples were homogenized using a 27 G needle, followed by centrifugation at $1000 \times g$, retrieval of the supernatant and IP with M2 magnetic beads (50% slurry) at 4 °C for 60 min. Post IP, beads were washed 4 times using wash buffer (20 mM HEPES; 150 mM KCl; 0.1% NP-40) followed by 50 mM HEPES. Immunoprecipitated proteins were eluted by incubating beads with SDS sample buffer containing 10% β-mercaptoethanol for 5 min at 95 °C, followed by western blotting.

## In vitro GST pull-down assay

GST and GST-tagged macroH2A1.1 macrodomains (GST-1.1) were expressed in BL21 E. coli bacteria grown O/N, at 18 °C. The soluble recombinant proteins were purified on Glutathione agarose beads (Thermo Fisher Scientific) by standard methods. Human full-length recombinant His-TOP1 protein (Sino Biological) was incubated O/N with an equal amount of GST or GST-1.1 and pre-cleared Glutathione beads, at 4 °C in binding buffer (50 mM Tris, pH = 7.5, 200 mM NaCl, and 0.02% NP40). Following co-immunoprecipitation, beads were extensively washed in the same buffer, and bound proteins were eluted in SDS sample buffer containing 10% β-mercaptoethanol for 5 min at 95 °C, followed by western blotting and detection of bound TOP1 by human anti-His-tag antibody (Abcam, Cat# ab18184) and GST/GST-1.1 by anti-GST antibody (Abcam, Cat# ab18184). For the enzymatic induction of ADP-ribosylation of TOP1 protein, recombinant His-TOP1 was incubated with an enzymatically active recombinant human PARP1 protein (Abcam), NAD$^+$ (Sigma) and "activated DNA" for PARP assays (BPS Bioscience) in the reaction buffer (50 mM Tris, pH = 8.0, 50 mM NaCl, 3 mM MgCl$_2$) for 30 min at the room temperature. Following enzymatic reaction, modified TOP1 protein was incubated with GST/GST-1.1 and beads for the GST pull-down, as described above.

## Cell viability assays

For clonogenic survival assays, cells were seeded at 300 cells per well in six-well Evap EDGE plates 8–16 h prior to treatment with the indicated doses of CPT with or without 0.1 µM Olaparib. After 72 h treatment, cells were released from drugs, maintained for 7–10 days, and then fixed with staining solution (6% glutaraldehyde and 0.5% crystal violet). Colonies were counted by Scan 4000 (Interscience). For MTT assays, cells were seeded at 2000–4000 cells per well in 96-well plates and treated with the indicated doses of CPT for 24 h. 48–72 h after the end of drug treatment, cells were treated with 0.5 mg/mL MTT reagent (Sigma Aldrich) for 2 h, washed and incubated with 100 µL DMSO for at least 20 min. Absorbance was measured at 570 nm using SpectraMax M5 (Molecular Devices), with SoftMax Pro 5.2 software (Molecular Devices). A modified MTT assay was performed to assess long-term cell survival in cell lines with impaired colony formation. 10 days after the end of drug treatment, cells were stained with 0.5 mg/mL MTT for 2 h, washed and incubated with 1-2 mL DMSO for at least 20 min. 100 µL of supernatant from each sample was transferred to 96-well plates and processed as above.

## RNA extraction and RT-PCR

Total RNA was extracted using the TRIzol™ reagent according to the manufacturer's instructions (Invitrogen). cDNA was synthesized from 2 µg of total RNA using PrimeScript RT Master Mix (Takara), and expression of the indicated genes was analyzed by quantitative RT-PCR using the CFX Opus 96 Real-Time PCR System (Bio-Rad) (see Supplementary Table 1 for primer sequences).

## Cleavage under targets & release using nuclease (CUT&RUN)

CUT&RUN was performed essentially as described with minor modifications[64]. Concanavalin A-coated (ConA-coated) bead slurry (Bangs Laboratories, cat. BP531) was activated in binding buffer (20 mM HEPES, pH 7.5; 10 mM KCl; 10 mM CaCl$_2$; 10 mM MnCl$_2$). $10^6$ cells were harvested and resuspended in 3 mL wash buffer (20 mM HEPES pH 7.5; 150 mM NaCl; 0.5 mM Spermidine; Complete Protease inhibitor (Sigma-Aldrich)) at RT. 10 µL of activated bead slurry was added, and the samples were rotated for 10 minutes at room temperature. Bead-bound cells were isolated on a magnet stand and resuspended in 300 µL antibody buffer (dig-Wash buffer: wash buffer plus 0.05% digitonin, supplemented with 2 mM EDTA). 0.3–0.5 µg of each primary antibody was added per sample and incubated under rotation at 4 °C overnight. Beads were washed and resuspended in 300 µL dig-Wash buffer. 5 µL pAG-MNase (EpiCypher) was added, and bead slurries were rotated at 4 °C for 1 h. Samples were washed twice in dig-Wash buffer and MNase was activated via addition of 2 mM CaCl$_2$, followed by a 1 h incubation at 0 °C. MNase was inactivated with 2× STOP Buffer (340 mM NaCl; 20 mM EDTA; 4 mM EGTA; 0.05% digitonin; 100 µg/mL RNase A; 50 µg/mL glycogen) mixed with 1–2 ng CUTANA E. coli Spike-in DNA (EpiCypher). CUT&RUN fragments were released via incubation at 37 °C for 30 minutes, separated on a magnet stand, and purified using the MiniElute PCR Purification Kit (Qiagen). Selected CUT&RUN samples were analyzed by qPCR enrichment using a CFX Opus 96 Real-time PCR System (Bio-rad), see Supplementary Table 1 for primer sequences.

## TOP1 covalent adduct detection coupled to NGS (TOP1 CAD-Seq)

For the detection of steady-state TOP1ccs, $10^7$ cells were pre-treated with MG132 (10 µM, Sigma-Aldrich) for 30 min, followed by a brief 3–5 min pulse of CPT (20 µM). MG132 was added to inhibit proteasomal degradation of TOP1ccs, which precedes TDP1-mediated debulking, to stabilize TOP1:DNA adducts. For detection of TOP1ccs after prolonged damage, cells were treated with 20 µM CPT for 30 min in the absence of MG132 to enhance TOP1cc trapping while simultaneously allowing to measure TOP1cc turnover. DNA isolation and TOP1 IP were performed essentially as described by Kuzin et al.[30]. Following treatments, cells were immediately lysed in the absence of crosslinking and briefly sonicated using a Fisher Scientific Sonic Dismembrator Model 100. Covalent adduct-containing genomic DNA was ethanol-precipitated and resuspended in TE-0.1% SDS plus 0.5 mM AEBSF (Sigma-Aldrich). Samples were further sonicated with a Covaris ME220 sonicator for

5 min using the High Cell protocol in 1 mL milliTUBEs to produce ~1 kb-sized fragments. For immunoprecipitation, 2 µg of anti-TOP1 antibody (Abcam) was preincubated with 25 µL of Protein A/G magnetic beads (Pierce) in RIPA buffer at 4 °C for 3 hours. DNA from $10^7$ cells was added to the Protein A/G–antibody complexes and incubated overnight at 4 °C. Magnetic beads were washed, DNA was eluted in 100 µL of TE-0.5% SDS buffer with Proteinase K at 60 °C for 4 hours and purified using the MiniElute PCR purification kit (Qiagen). Library generation and NGS were performed as described below. For CAD at the MS2 locus, eluted DNA was subjected to qPCR using a TSS-proximal primer pair and a second primer pair in the CMV enhancer (see Supplementary Table 1).

### Library preparation and next-generation sequencing

NGS libraries were prepared using the ThruPLEX DNA-Seq Kit (Takara Bio USA, cat. R400675), following the manufacturer's instructions. 1–10 ng DNA was used per sample and PCR cycle numbers were adjusted as recommended. Samples were bar-coded using the DNA HT Dual Index Kit (Takara Bio USA).

Amplified libraries were purified using AMPure XP beads (Beckman Coulter) at a 1:1 (v/v) ratio. Library fragment size and concentration were measured using TapeStation (Agilent D1000 reagent). Up to 18 samples were pooled at a final concentration of 4 nM per sample. Paired-end Illumina sequencing was performed on the NextSeq 2000 Sequencing System.

### NGS data analysis

The quality of the raw sequenced reads was examined with FastQC. For CUT&RUN, adapters were trimmed using TrimGalore in paired-end mode. For CAD-Seq, reads were trimmed using cutadapt followed by read pairing with fastq-pair. PE reads were aligned against the hg38 human reference genome using Bowtie2[65]. Aligned reads were sorted, indexed and deduplicated using Picard (Broad Institute, GitHub) and Samtools[66]. For CUT&RUN, reads were further aligned against the E. coli (K12_DH10B) genome and human-aligned reads were normalized to CUTANA E. coli spike-in content following EpiCypher's instructions. When indicated, biological replicates were averaged using the *bigwigAverage* tool in deepTools[67]. For CAD-Seq, replicates were merged using Samtools to increase read depth. Fastq and bam files are available through dbGaP (phs003729). H3K27me3 (GSM949581) and PARP1 ChIP-Seq data (GSM1517306) were obtained from Gene Expression Omnibus (GEO)[26,28]. Reads mapping to ENCODE blacklist regions were excluded from downstream analyses[68].

### Peak-calling.

FLAG-macroH2A1.1 broad peaks were called using SICER[69] with 200 bp window size, 150 fragment size, and 600 bp gap size. TOP1 narrow peaks were called using MACS2[70]. The R package ChIP-seeker was used to annotate peaks based on genomic context[71]. TSS-proximal TOP1 peaks were defined as all TOP1 peaks within ±3000 bp of the nearest TSS. TSS-distal (Non-TSS) TOP1 peaks were defined as all TOP1 peaks greater than 10 kb away from the nearest TSS, strand orientation was considered only for TSS-proximal peaks.

### Z-Normalization.

To standardize the genomic signal across different TOP1 CAD-Seq data sets, we performed Z-normalization on BigWig files, where indicated. Each score in the BigWig file was transformed into a Z-score by subtracting the mean and dividing by the standard deviation (SD): Z-score = (score−mean)/SD.

### Correlation analysis.

Similarity between CUT&RUN data sets was evaluated using the *multiBigwigSummary* and *plotCorrelation* features in deepTools. The *multiBigwigSummary* tool was employed to compute average scores across individual bigwig files for different experimental replicates based on equally sized 10 kb bins, or the indicated genomic regions in bed format, *plotCorrelation* was used to compute and visualize the Spearman correlation coefficients between these datasets.

### Jaccard Index analysis.

The Jaccard Index provides a measure of similarity between peak sets that is comparable across sets of varying size. Jaccard Indices were calculated using bedtools[72], and are defined as the bases in the intersection divided by the bases in the union, producing values bounded by zero and 1 (no overlap and complete overlap, respectively). To determine whether this metric was different than expected by chance, permuted Jaccard values were calculated based on random shuffles of both peak sets ($n = 1000$ shuffles) using the shuffle command from bedtools within the bounds of the human genome assembly hg38. All peak sets were called with SICER[69].

### Profile plots and heatmaps.

The *computeMatrix* command from deepTools was used to calculate scores per genome regions defined in a BED file. Heatmaps were generated using the *plotHeatmap* command, profile plots using the *plotProfile* command. TSS separation based on gene expression quartiles was based on RNA-Seq data from[21].

### LOESS smoothing of average profiles.

To smooth the average profiles of our genomic data, we applied LOESS (Locally Estimated Scatterplot Smoothing) using the *geom_smooth* function in R's ggplot2 package. We utilized span = 0.15 to capture localized trends in the data while avoiding overfitting, a span = 0.025 was used for Fig. 2b to provide finer detail.

### Kaplan–Meier analyses.

Correlation analyses between macroH2A1.1 expression and overall patient survival were performed using the Ovarian Cancer data set in the Kaplan–Meier Plotter survival analysis tool[46,47] and the macroH2A1.1-specific Affy probe set 214500_at ref. [19]. Patient subsets and cutoffs are as indicated in the figure legend, array quality control was set to "exclude biased arrays".

### Reporting summary

Further information on research design is available in the Nature Portfolio Reporting Summary linked to this article.

## Data availability

The genomic data generated in this study have been deposited in the database of Genotypes and Phenotypes (dbGaP) under accession code phs003729. According to the NIH Genomic Data Sharing Policy, the large human cell line-derived genomic data set generated in this study is subject to JHU Institutional Certification, which requires controlled data access. The genomic data are available for health/medical/biomedical research use. To access the data, login to the dbGaP controlled-access portal to initiate a project. Source data are provided with this paper.

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

## Acknowledgements

We thank Yves Pommier for critical reading of this manuscript, Yan Zhang and Ludmila Danilova (JHU Experimental Cancer Genomics Core) for help with mutation signature analyses, Michael Matunis and Allison DeHaas (JHU Bloomberg School for Public Health) for microscope assistance, Anne-Claire Lavigne and Kerstin Bystricky (University of Toulouse) for MDA-MB231 RNA-Seq data, and Roger Greenberg (University of Pennsylvania), Yves Pommier (NCI), Daniele Gilkes (JHU) and Tian-Li Wang (JHU) for reagents. This work was supported by internal Johns Hopkins University funding and the National Institutes of Health under award numbers R35GM153484, P50CA228991 (P.O.) and R01CA285725 (P.O. and T.H.L.), by Knut och Alice Wallenbergs Stiftelse: KAW 2016.0161, KAW 2022.0380, and KAW 2022.0189, Vetenskapsrådet: 2021-02630 VR, Cancerfonden: 21 1771 Pj01 H, and Karolinska Institutet Consolidator: 2-190/2022 (L.H. and V.K.), and the national grant PID2021-126907NB-I00 from MCIN/AEI/10.13039/501100011033, co-funded by European Regional Development Fund (M.B., M.F., and D.C.), and the postdoctoral fellowship HORIZON-MSCA-2022-PF-01-101108823 (M.F.). Computations and data storage were supported by the Advanced Research Computing at Hopkins (ARCH) core facility (rockfish.jhu.edu), funded in part by National Science Foundation (NSF) grant number OAC1920103, and by the National Academic Infrastructure for Supercomputing in Sweden resources at UPPMAX (NAISS projects 2023/23-603 and 2023/22-258), with partial funding from the Swedish Research Council (grant 2022-06725).

## Author contributions

P.O. and T.H.L. conceived the study. T.H.L., P.O., and C.X.Q. designed the experiments. T.H.L. performed all cell-based and biochemistry assays. C.X.Q., Z.Z., and T.W. performed sequencing experiments, C.X.Q., V.K., and Z.Z. analyzed genomics data with help from L.B. Y.S., and T.G. performed MS2 reporter cell assays with help from T.H.L., M.F. performed in vitro biochemistry, V.R. and X.Z. generated CRISPR knockout cells. D.C. and M.B. provided macroH2A1 expression vectors. P.O. and T.H.L. wrote the manuscript with input from L.B., M.B., and C.X.Q.

## Competing interests

The authors declare no competing interests.
