## [Transparent Peer Review file · Nature Communications]

Epigenetic control of Topoisomerase 1 activity presents a cancer vulnerability

Corresponding Author: Dr Philipp Oberdoerffer

Version 0:

Reviewer comments:

Reviewer #1

(Remarks to the Author)

Lee et al investigate how the poly-ADP-ribose (PAR)-binding histone variant macroH2A1.1 influences repair of cytotoxic topoisomerase 1 (TOP1) DNA-protein crosslinks (DPCs). In a first instance, the authors demonstrate that macroH2A1.1 and TOP1 reside in the same chromosomal regions. ColP experiments further indicate that the association between the two factors increases when TOP1-DPCs are induced by camptothecin. The authors proceed by showing that depletion of macroH2A1.1 slightly decreases clearance of TOP1ccs which is suppressed by re-expression of the WT protein but not of a variant that cannot bind PAR. The authors further show that macroH2A1.1 helps to recruit the single-strand break (SSB) repair protein XRCC1 in a PAR dependent manner and that in the absence of macroH2A1.1, SSBs accumulate. Following these observations, it is shown that XRCC1 and TOP1 accumulate at a transcriptional reporter, which the authors attribute to TOP1-DPC formation and subsequent recruitment of XRCC1. The recruitment of XRCC1, but not of TOP1, was reduced in the absence of macroH2A1.1. In addition, the authors convincingly show that macroH2A1.1 expression predicts the response to several TOP1-DPC inducing drugs across a panel of cancer cell lines. Finally, the authors correlate a mutational cancer signature that is linked to base excision mediated repair of deaminated cytosines (but not to TOP1-DPCs) to macroH2A1.1 levels and in a second step to the success of chemotherapies with TOP1-DPC-inducing drugs.

While the effect of macroH2A1.1 on TOP1-DPC clearance and cytotoxicity was clearly shown by the authors, I am less convinced by the proposed underlying mechanism. The discussion of TOP1-DPC repair is simplistic and does not reflect the current state of the literature. Thus, major revisions are needed before I can recommend publication.

Specific points:

1. It remains entirely unclear which step of TOP1-DPC repair is promoted by macroH2A1.1. TOP1-DPC repair entails the degradation of the crosslinked topoisomerase followed by removal of the remaining peptide adduct by TDP1, generating a clean 3' end. In the final step, the SSB is repaired by the SSB repair machinery (and not BER as often stated in the manuscript). The fact that DPC clearance is delayed upon depletion of macroH2A1.1 suggests that it is not simply the repair of the single-strand break but the debulking of the protein adduct that is affected. Indeed, a recent study revealed that the proteasomal degradation of TOP1-DPCs occurs in a PAR-dependent manner (Fabian et al, Nat Commun, 2024). Also, TDP1 is regulated by PARP1 and PAR during TOP1-DPC repair (e.g., Das et al NAR, 2014). These studies are neither discussed, cited or incorporated into the author's model. Thus, it remains unclear how macroH2A1.1 promotes repair of TOP1-DPCs.
2. Figure 2B: I am not an expert, but the Top1cc signal appears to resemble the pattern observed in ChIP-Seq input samples with reduced coverage for the TSS region that tends to be preferentially sheared during sonication. Are the authors confident that this is a true TOP1-DPC signal and not simply background?
3. Figure 2E: I wonder whether these small effect sizes are biologically meaningful.
4. Figure 5: It remains unclear to me whether any of the of the observed effects are caused by TOP1-DPC formation. The data are correlative at best. Can the authors use their TOP1-CAD-Seq/qPCR protocol, to confirm that there is any TOP1-DPC induction at the reporter site?

5. Figure 7A: The correlation of the TOP1-independent SBS2 signature with macroH2A1.1 expression is not related to the topic of the manuscript. I suggest removing this analysis.

Minor points

1. Figure 2C: Please provide the actual profiles as in 2B and not just the ratios between ctrl and macroH2A1.1 depletion.
2. TOP1-DPCs are introduced as a single-stranded DNA lesion. This definition does not reflect the complex nature of the lesion that entails the covalent attachment of a large protein adduct.
3. In the introduction, the authors states that PARP1's role in TOP1-DPC repair is the sensing of the SSB arising during TOP1-DPC formation (the provided citations do not support this point). PARP1 cannot sense the nick unless TOP1 is degraded first. Thus, if PARP1 is important for clearance of the entire DPC, its role must go beyond sensing of the nick (see also major point 1)

Reviewer #2

(Remarks to the Author)

In the study by Lee et al, the authors discovered how cells are protected from aberrant TOP1 activity at topological stress hot spots. They found that the histone variant macroH2A1.1, but not the alternatively spliced macroH2A1.2 isoform, through its ability at binding poly(ADP-ribose) (PAR), establishes a permissive chromatin context to facilitate TOP1 cleavage complexes (TOP1cc) resolution. Furthermore, the authors show mechanistically that macroH2A1.1 facilitates PAR-dependent recruitment of the TOP1cc repair effector XRCC1 to protect from ssDNA damage, explaining the protective of macroH2A1.1 on TOP1cc induced DNA damage. Importantly they additionally show that cancers with impaired macroH2A1.1 splicing show increased sensitivity to TOP1 poisons and therefore low macroH2A1.1 expression correlated with improved survival in cancer patients treated with TOP1 inhibitors.

This is a significant work to the genome topology and DNA repair field with potential clinical interest, showcasing that macroH2A1 alternative splicing serves as an epigenetic modulator of TOP1-associated genome maintenance and as potential cancer vulnerability. The manuscript is very well written and the presented data are solid and convincing. Figures are presented in a clear manner and the methodology used is sound, although in few cases the study could be improved by few genome-wide experiments (see comments). I would suggest that the study is ready for publication in Nat Commun, but could be further improved if the authors address the minor comments found below.

- The authors assessed XRCC1 recruitment to CPT-induced DNA lesions in the presence or absence of macroH2A1.1 (as shown by immunofluorescence in MCF7 cells). These experiments could be expanded to provide additional information on genomic distribution (e.g. whether XRCC1 recruitment is found only around TSSs or dependencies on nascent transcription etc) by profiling XRCC1 occupancy genome wide by performing ChIP-seq or CUT&RUN methodologies under similar conditions. These experiments will tremendously support the existing findings and expand them further.
- The authors write: "TOP1-associated macroH2A1.1 domains (TMDs) were notably distinct from the well-characterized, heterochromatin-associated macroH2A1.1 domains, as they were depleted for H3K27me3 (Fig. 1C). We thus propose that TMDs present a unique macroH2A1.1 chromatin environment reminiscent of macroH2A1 regions previously associated with PARP-dependent gene regulation". Later in the ms the authors say: "Of note, macroH2A1.1 loss-associated TOP1cc accumulation extended into the flanking regions of macroH2A1.1 high TOP1 peaks, consistent with abundant macroH2A1.1 enrichment beyond the TOP1 peak at these sites (Fig. 2E)".
Introducing a new term (TMDs) to describe regions in chromatin where macroH2A1.1 and TOP1 just partly colocalise (as other hundreds of proteins) in chromatin is an overkill. Please, revise the relevant sections.
- The authors report that they tested the physical association of macroH2A1.1 and TOP1 by Co-IP. This is clearly not a direct interaction as it is sensitive to PARP inhibitors. If the authors want to claim direct interactions, they should show interactions at the level of recombinant proteins. Characterizing the complex further (other proteins, which domains are involved etc) would further improve the study.
- Page 14, title of Figure legend. MacroH2A1.1 instead of MacoH2A1.1
- Page 6, first paragraph the reference 19 is included twice.

Reviewer #3

(Remarks to the Author)

In this manuscript, the authors identify that the histone variant macroH2A1.1, through its binding to poly(ADP)-ribose (PAR), binds to topoisomerase I (TOP1), as well as to the TOP1 cleavage complex (TOP1cc) repair proteins PARP1 and XRCC1, and promotes TOP1cc removal, particularly at sites of active transcription. They also show that low expression of macroH2A1.1 is predictive of increased sensitivity of cancer cells to camptothecin (CPT) and improved survival of cancer patients treated with CPT derivatives. This study, which clearly establishes a connection between epigenetics and TOP1 activity, significantly deepens our understanding of the mechanisms of TOP1cc removal. My comments are indicated below, in particular to clarify whether macroH2A1.1 promotes TOP1cc removal through the TDP1 pathway.

1. The functional interaction of macroH2A1.1 with PARP1 and XRCC1 suggests that macroH2A1.1 engages the TDP1 pathway for TOP1cc removal. That should be tested as PARP1 and XRCC1 have other functions besides removing TOP1cc. Does macroH2A1.1 interacts with TDP1, which is also known to be PARylated? Functionally, it should be also

determined whether macroH2A1.1 and TDP1 act in the same pathway for TOP1cc removal. That could be tested by assessing TOP1 lesions induced by CPT (e.g., by alkaline comet assay) under single vs concurrent depletion of macroH2A1.1 and TDP1.

2. One consequence of the defective removal of TOP1cc at transcription sites is the formation of DNA double-strand breaks (DSBs). The author should therefore test whether depletion of macroH2A1.1 promotes the formation of transcription-dependent DSBs in response to CPT, e.g., by assessing γ H2AX foci in EdU-negative vs EdU-positive cells.

3. The design of the experiment in Fig 2C (steady state vs. damage TOP1cc) should be better explained and the levels of TOP1cc shown between these two conditions, e.g., by RADAR assay.

4. It is unclear why depletion of macroH2A1.1 does not increase TOP1cc in response to CPT (Fig 2F), whereas it does increase Olive tail moment (Fig 4E).

Version 1:

Reviewer comments:

Reviewer #1

(Remarks to the Author)

The authors have addressed all my concerns.

Reviewer #2

(Remarks to the Author)

The authors have adequately revised the manuscript which is now suitable for publication to Nat Commun.

Reviewer #3

(Remarks to the Author)

The authors have adequately addressed all my comments and concerns by conducting several additional experiments and providing new data that support and extend their original conclusions. The revised manuscript is improved and is of significant interest to a broad scientific audience.

Response to the reviewers

We would like to thank the reviewers for their overall positive evaluation, helpful comments and constructive suggestions. We have now addressed their concerns and added substantial textual and experimental revisions, which we believe significantly strengthen our conclusions. Specifically, we have added analyses dissecting the effect of macroH2A1.1 on TDP1 function during TOP1cc repair, which further clarifies the role of macroH2A1.1 in TOP1cc resolution. We also provide evidence for increased DNA double-strand break formation as a result of unrepaired TOP1ccs in macroH2A1.1-deficient cells, underlining the genotoxic impact of macroH2A1.1 loss. Finally, we show that macroH2A1.1 depletion results in XRCC1 loss at TOP1 peaks and transcription start sites genome-wide, extending our findings from the MS2 transcription reporter locus to endogenous, transcribed loci. Additional experiments and clarifications have been added to corroborate and extend our findings, all major textual changes are highlighted in blue. We hope that in light of these revisions, you now consider this manuscript suitable for Nature Communications. A point by point response to individual reviewer comments follows below.

Reviewer #1:

Lee et al investigate how the poly-ADP-ribose (PAR)-binding histone variant macroH2A1.1 influences repair of cytotoxic topoisomerase 1 (TOP1) DNA-protein crosslinks (DPCs). In a first instance, the authors demonstrate that macroH2A1.1 and TOP1 reside in the same chromosomal regions. ColP experiments further indicate that the association between the two factors increases when TOP1-DPCs are induced by camptothecin. The authors proceed by showing that depletion of macroH2A1.1 slightly decreases clearance of TOP1ccs which is suppressed by re-expression of the WT protein but not of a variant that cannot bind PAR. The authors further show that macroH2A1.1 helps to recruit the single-strand break (SSB) repair protein XRCC1 in a PAR dependent manner and that in the absence of macroH2A1.1, SSBs accumulate. Following these observations, it is shown that XRCC1 and TOP1 accumulate at a transcriptional reporter, which the authors attribute to TOP1-DPC formation and subsequent recruitment of XRCC1. The recruitment of XRCC1, but not of TOP1, was reduced in the absence of macroH2A1.1. In addition, the authors convincingly show that macroH2A1.1 expression predicts the response to several TOP1-DPC inducing drugs across a panel of cancer cell lines. Finally, the authors correlate a mutational cancer signature that is linked to base excision mediated repair of deaminated cytosines (but not to TOP1-DPCs) to macroH2A1.1 levels and in a second step to the success of chemotherapies with TOP1-DPC-inducing drugs.

While the effect of macroH2A1.1 on TOP1-DPC clearance and cytotoxicity was clearly shown by the authors, I am less convinced by the proposed underlying mechanism. The discussion of TOP1-DPC repair is simplistic and does not reflect the current state of the literature. Thus, major revisions are needed before I can recommend publication.

We appreciate the overall positive evaluation, constructive feedback and valuable suggestions. We have now revised the manuscript as outlined below to provide a better understanding of how macroH2A1.1 functions in TOP1-DPC clearance.

Specific points:

1. It remains entirely unclear which step of TOP1-DPC repair is promoted by macroH2A1.1. TOP1-DPC repair entails the degradation of the crosslinked topoisomerase followed by removal of the remaining peptide adduct by TDP1, generating a clean 3' end. In the final step, the SSB is repaired by the SSB repair machinery (and not BER as often stated in the manuscript). The fact that DPC clearance is delayed upon depletion of macroH2A1.1 suggests that it is not simply the repair of the single-strand break but the debulking of the protein adduct that is affected. Indeed, a recent study revealed that the proteasomal degradation of TOP1-DPCs occurs in a PAR-dependent manner (Fabian et al, Nat Commun, 2024). Also, TDP1 is regulated by PARP1 and PAR during TOP1-DPC repair (e.g., Das et al NAR, 2014). These studies are neither discussed, cited or incorporated into the author's model. Thus, it remains unclear how macroH2A1.1 promotes repair of TOP1-DPCs.

We thank the reviewer for this comment and have revised the introduction to include a more detailed discussion of the role of PARP1 in regulating TOP1 repair, TOP1 DPC degradation and TDP1 activity. We have further corrected BER to SSB throughout the manuscript.

*We agree with the reviewer that our RADAR and CAD-Seq data point to a role for macroH2A1.1 in TOP1cc debulking. To experimentally dissect the impact of macroH2A1.1 loss on this process, we have now assessed TDP1 recruitment to chromatin in response to CPT-induced damage. We find that macroH2A1.1 depletion impairs TDP1 accumulation in a manner similar to PARP inhibition, and that macroH2A1.1 and TDP1 act in the same pathway to facilitate the recruitment of the downstream repair factor XRCC1 to damaged chromatin (**new Fig. 4e**). Consistent with the latter, macroH2A1.1 depletion did not further aggravate DNA break formation following TDP1 loss (**new Fig. S7b**). Together, these findings provide a compelling rationale for macroH2A1.1 as a coordinator of PAR-dependent TOP1cc repair events, from TOP1cc debulking to ssDNA break repair. We also note that, in addition to PARP1, XRCC1 was shown to enhance TDP1 activity (Plo et al, 2003). XRCC1 recruitment via macroH2A1.1 may, therefore, at least in part contribute to the effect of macroH2A1.1 on TOP1cc debulking. The results and discussion have been expanded accordingly.*

2. Figure 2B: I am not an expert, but the Top1cc signal appears to resemble the pattern observed in CHIP-Seq input samples with reduced coverage for the TSS region that tends to be preferentially sheared during sonication. Are the authors confident that this is a true TOP1-DPC signal and not simply background?

We have revised Figure 2B to show a representative TOP1 CAD-Seq experiment, for which an accompanying input control was sequenced. Robust TOP1cc accumulation is observed in the TSS-flanking regions compared to the background signal detected in the input sample. Moreover, relative TOP1cc depletion at the TSS compared to the surrounding DNA is more pronounced in the IP sample, in agreement with our previous work in other cell lines (Baranello et al, 2016; Das et al, 2022). Please also note the absence of reduced TSS coverage upon macroH2A1.1 depletion (see new Fig. S2d). Together, these findings show that TSS-associated TOP1cc loss is not merely the result of preferential sonication.

3. Figure 2E: I wonder whether these small effect sizes are biologically meaningful.

We thank the reviewer for the opportunity to clarify this point. Please note that the TOP1 CAD-seq assay captures TOP1-DNA adducts in the absence of formaldehyde crosslinking. TOP1 CAD-Seq, thus, requires high CPT concentrations (20 μ M) for robust TOP1cc detection, which results in relatively limited TOP1cc turnover within the 30-minute time frame measured. Despite these technical limitations, which we have now discussed in the results section (page 5), we consider the CAD-Seq data an important addition as they provide the most readily available approach for the genomic mapping of TOP1cc lesion turnover. Please note that our CAD-Seq findings are corroborated by kinetic analyses of TOP1cc turnover using the RADAR assay, which is compatible with significantly lower CPT concentrations and shows a robust TOP1cc repair defect. The biological relevance of the effect on TOP1cc repair is further illustrated by increased DNA break formation and impaired cell survival upon CPT treatment in the absence of macroH2A1.1 (Figs. 6, 7).

4. Figure 5: It remains unclear to me whether any of the of the observed effects are caused by TOP1-DPC formation. The data are correlative at best. Can the authors use their TOP1-CAD-Seq/qPCR protocol, to confirm that there is any TOP1-DPC induction at the reporter site?

We thank the reviewer for this suggestion and have now performed Covalent Adduct Detection (CAD) followed by qPCR in our MS2 reporter system. We observe pronounced, Dox-induced accumulation of TOP1cc signal near the MS2 TSS, demonstrating that MS2 induction promotes TOP1:DNA adducts at this locus (new Fig.5c).

5. Figure 7A: The correlation of the TOP1-independent SBS2 signature with macroH2A1.1 expression is not related to the topic of the manuscript. I suggest removing this analysis.

We agree with the reviewer and have removed this panel.

Minor points

1. Figure 2C: Please provide the actual profiles as in 2B and not just the ratios between ctrl and macroH2A1.1 depletion.

We have now added TOP1 CAD-Seq profiles for “steady state” and “damage” in control and knockdown cells in new Fig. S2d. We stratified the data by high and low RNA Seq expression quartiles to reflect the dependence of the TOP1cc signal on gene expression levels.

2. TOP1-DPCs are introduced as a single-stranded DNA lesion. This definition does not reflect the complex nature of the lesion that entails the covalent attachment of a large protein adduct.

We thank the reviewer for this comment and have expanded on the complex nature of TOP1cc lesions and repair in paragraph 2 of our revised introduction as follows: “Given their complex nature, the resolution of TOP1ccs requires specialized TOP1:DNA adduct removal before the resulting ssDNA gap can be repaired via canonical single-strand break repair (SSBR).”

3. In the introduction, the authors states that PARP1's role in TOP1-DPC repair is the sensing of the SSB arising during TOP1-DPC formation (the provided citations do not support this point). PARP1 cannot sense the nick unless TOP1 is degraded first. Thus, if PARP1 is important for clearance of the entire DPC, its role must go beyond sensing of the nick (see also major point 1)

We have now elaborated on the multifaceted, known roles of PARP during TOP1cc repair, including recent work demonstrating PARP1 activation by DPCs adjacent to ssDNA gaps (Fabian et al, 2024), its role in TDP1 activation and its tightly regulated effect on DPC degradation (Das et al, 2014; Fabian et al., 2024; Sun et al, 2021). See revised introduction (page 2) and discussion (page 12).

Reviewer #2 (Remarks to the Author):

In the study by Lee et al, the authors discovered how cells are protected from aberrant TOP1 activity at topological stress hot spots. They found that the histone variant macroH2A1.1, but not the alternatively spliced macroH2A1.2 isoform, through its ability at binding poly(ADP-ribose) (PAR), establishes a permissive chromatin context to facilitate TOP1 cleavage complexes (TOP1cc) resolution. Furthermore, the authors show mechanistically that macroH2A1.1 facilitates PAR-dependent recruitment of the TOP1cc repair effector XRCC1 to protect from ssDNA damage, explaining the protective of macroH2A1.1 on TOP1cc induced DNA damage. Importantly they additionally show that cancers with impaired macroH2A1.1 splicing show increased sensitivity to TOP1 poisons and therefore low macroH2A1.1 expression correlated with improved survival in cancer patients treated with TOP1 inhibitors.

This is a significant work to the genome topology and DNA repair field with potential clinical interest, showcasing that macroH2A1 alternative splicing serves as an epigenetic modulator of TOP1-associated genome maintenance and as potential cancer vulnerability. The manuscript is very well written and the presented data are solid and convincing. Figures are presented in a clear manner and the methodology used is sound, although in few cases the study could be improved by few genome-wide experiments (see comments). I would suggest that the study is ready for publication in Nat Commun, but could be further improved if the authors address the minor comments found below.

We thank the reviewer for the positive feedback and have further improved our manuscript following the suggestions below.

- The authors assessed XRCC1 recruitment to CPT-induced DNA lesions in the presence or absence of macroH2A1.1 (as shown by immunofluorescence in MCF7 cells). These experiments could be expanded to provide additional information on genomic distribution (e.g. whether XRCC1 recruitment is found only around TSSs or dependencies on nascent transcription etc) by profiling XRCC1 occupancy genome wide by performing ChIP-seq or CUT&RUN methodologies under similar conditions. These experiments will tremendously support the existing findings and expand them further.

We thank the reviewer for this suggestion. To complement our co-IP and IF-based studies and further assess the impact of macroH2A1.1 on XRCC1 association with chromatin, we have now performed CUT& RUN as well as analyses of chromatin-bound XRCC1 in the presence or absence

of macroH2A1.1 in response to CPT treatment. Our chromatin fractionation analyses corroborate that CPT-induced XRCC1 recruitment to chromatin is promoted by macroH2A1.1 as well as PARP activity (new Fig. 4e). These findings are further supported by XRCC1 CUT&RUN NGS, which revealed reduced XRCC1 recruitment upon macroH2A1.1 knockdown both at the TSS and at TOP1 peaks genome-wide, consistent with a recruitment defect localized to sites of TOP1 activity (new Fig. 4f, g, new Fig. S5).

- The authors write: “TOP1-associated macroH2A1.1 domains (TMDs) were notably distinct from the well-characterized, heterochromatin-associated macroH2A1.1 domains, as they were depleted for H3K27me3 (Fig. 1C). We thus propose that TMDs present a unique macroH2A1.1 chromatin environment reminiscent of macroH2A1 regions previously associated with PARP-dependent gene regulation”. Later in the ms the authors say: “Of note, macroH2A1.1 loss-associated TOP1cc accumulation extended into the flanking regions of macroH2A1.1 high TOP1 peaks, consistent with abundant macroH2A1.1 enrichment beyond the TOP1 peak at these sites (Fig. 2E)”.

Introducing a new term (TMDs) to describe regions in chromatin where macroH2A1.1 and TOP1 just partly colocalise (as other hundreds of proteins) in chromatin is an overkill. Please, revise the relevant sections.

We apologize for any confusion this term may have caused. Our goal was to emphasize that TOP1-associated macroH2A1.1 domains are distinct from macroH2A1.1 regions associated with repressive chromatin. However, we agree that macroH2A1.1 is likely not the only defining feature of these regions and have removed the acronym “TMD” as suggested (see page 4).

- The authors report that they tested the physical association of macroH2A1.1 and TOP1 by Co-IP. This is clearly not a direct interaction as it is sensitive to PARP inhibitors. If the authors want to claim direct interactions, they should show interactions at the level of recombinant proteins. Characterizing the complex further (other proteins, which domains are involved etc) would further improve the study.

We thank the reviewer for the opportunity to further clarify this important point. We have now performed in vitro binding assays between the macroH2A1.1 macro-domain and purified TOP1 protein in the presence or absence of active PARP1. These assays confirm that the TOP1:macro-domain interaction is dependent on PARP1 activity (new Fig. S1c), in agreement with our cell-based findings using PARP inhibitors or the PAR-binding domain-deficient macroH2A1.1 G224E mutant. We summarize these findings in our revised discussion, emphasizing that the observed macroH2A1.1 interactions may result from macroH2A1.1 PAR-binding domain association with PARylated protein, or that they “may involve PARylated intermediary factors. For example, auto-PARylated PARP1 may serve as a bridge between macroH2A1.1 nucleosomes and repair factors with conserved PAR binding motifs such as XRCC1.” (Page 12).

- Page 14, title of Figure legend. MacroH2A1.1 instead of MacoH2A1.1
- Page 6, first paragraph the reference 19 is included twice.

These errors have been corrected.

Reviewer #3 (Remarks to the Author):

In this manuscript, the authors identify that the histone variant macroH2A1.1, through its binding to poly(ADP)-ribose (PAR), binds to topoisomerase I (TOP1), as well as to the TOP1 cleavage complex (TOP1cc) repair proteins PARP1 and XRCC1, and promotes TOP1cc removal, particularly at sites of active transcription. They also show that low expression of macroH2A1.1 is predictive of increased sensitivity of cancer cells to camptothecin (CPT) and improved survival of cancer patients treated with CPT derivatives. This study, which clearly establishes a connection between epigenetics and TOP1 activity, significantly deepens our understanding of the mechanisms of TOP1cc removal. My comments are indicated below, in particular to clarify whether macroH2A1.1 promotes TOP1cc removal through the TDP1 pathway.

We thank this reviewer for the positive assessment of our work and have revised the manuscript to further explore the relationship between macroH2A1.1, TOP1cc turnover and TDP1.

1. The functional interaction of macroH2A1.1 with PARP1 and XRCC1 suggests that macroH2A1.1 engages the TDP1 pathway for TOP1cc removal. That should be tested as PARP1 and XRCC1 have other functions besides removing TOP1cc. Does macroH2A1.1 interact with TDP1, which is also known to be PARylated? Functionally, it should be also determined whether macroH2A1.1 and TDP1 act in the same pathway for TOP1cc removal. That could be tested by assessing TOP1 lesions induced by CPT (e.g., by alkaline comet assay) under single vs concurrent depletion of macroH2A1.1 and TDP1.

*We thank this reviewer and reviewer 1 for their insightful comments related to the TDP1 pathway. To address if macroH2A1.1 contributes to TDP1 function in TOP1cc repair, we have now tested the effect of macroH2A1.1 loss on TDP1 association with chromatin in the presence or absence of CPT treatment and PARPi. We find that chromatin-bound TDP1 increases upon CPT treatment in a PARP activity-dependent manner, which is significantly reduced in macroH2A1.1-depleted cells (**new Fig. 4e**). To determine if macroH2A1.1 and TDP1 function in the same pathway, we assessed the impact of combined loss of macroH2A1.1 and TDP1 on XRCC1 recruitment using the same assay. Inactivation of either protein impaired CPT-induced XRCC1 recruitment to chromatin and no additive effect was observed upon macroH2A1.1/TDP1 co-depletion (**new Fig. 4e**). Following this reviewer's suggestion, we further determined the impact of macroH2A1.1 depletion on DNA break formation in the presence or absence of TDP1 using the Comet assay. While loss of either protein alone increased DNA break frequency, co-depletion of both proteins did not further aggravate the effect of TDP1 loss (**new Fig. S7b**). Together, these findings support an epistatic relationship between macroH2A1.1 and TDP1 in TOP1cc repair. Of note, we were unable to detect a splice isoform-specific interaction between macroH2A1.1 and TDP1 in Co-IP analyses (**new Fig. S4h**), suggesting that the effect of macroH2A1.1 on TDP1 recruitment to chromatin is an indirect consequence of XRCC1 and/or PARP1 association with macroH2A1.1.*

2. One consequence of the defective removal of TOP1cc at transcription sites is the formation of DNA double-strand breaks (DSBs). The author should therefore test whether depletion of macroH2A1.1 promotes the formation of transcription-dependent DSBs in response to CPT, e.g., by assessing γ H2AX foci in EdU-negative vs EdU-positive cells.

We thank the reviewer for this valuable suggestion and have now included the proposed analyses of CPT-induced γ H2AX foci (new Fig 6b, Fig. S7c, d). While few DSBs were observed immediately after CPT treatment in EdU-negative cells, we detected a robust increase in DSB frequency following release from CPT specifically when macroH2A1.1 was absent. This observation is consistent with a progressive conversion of unresolved TOP1ccs and/or ssDNA breaks into DSBs, as has been previously observed due to collisions between TOP1ccs and the transcription machinery. Reconstitution of macroH2A1.1 KO cells with WT macroH2A1.1 but not the PAR-binding-deficient G224E mutant efficiently suppressed the DSB accumulation in EdU⁻ cells. Protection from S phase-associated DSBs in EdU⁺ cells was less pronounced, pointing to additional, macroH2A1.1-independent pathways that repair TOP1ccs associated with DNA replication (see page 9).

3. The design of the experiment in Fig 2C (steady state vs. damage TOP1cc) should be better explained and the levels of TOP1cc shown between these two conditions, e.g., by RADAR assay.

We have revised the legend for Fig. 2C to better explain the experimental design. We have further assessed TOP1cc levels for the two CAD-Seq conditions by RADAR assay (new Fig. S2e). Please note that the high CPT concentrations (20 μ M) required for efficient TOP1cc detection using TOP1 CAD-Seq results in globally elevated levels of TOP1cc, which in turn leads to relatively limited TOP1cc turnover (Fig. 2c-e, Fig. S2e). Despite these technical limitations, we consider the CAD-Seq data an important addition as TOP1 CAD-Seq provides the most readily available approach for the genomic mapping of TOP1cc lesion turnover.

4. It is unclear why depletion of macroH2A1.1 does not increase TOP1cc in response to CPT (Fig 2F), whereas it does increase Olive tail moment (Fig 4E).

We thank the reviewer for the opportunity to clarify. We attribute this observation to the extended treatment time used for ssDNA break detection in the Comet assay (1h, versus 30 min for RADAR assay) and a concomitant increase in the time available to repair TOP1ccs when macroH2A1.1 is present. This is consistent with previous work by the Pommier lab (Sun et al., 2021), which reported a decrease in TOP1cc accumulation after 1 h compared to 30 min of CPT treatment. For added clarity, we have now specified treatment times in the Figure legends in addition to the methods section.

References cited:

Baranello L, Wojtowicz D, Cui K, Devaiah BN, Chung HJ, Chan-Salis KY, Guha R, Wilson K, Zhang X, Zhang H et al (2016) RNA Polymerase II Regulates Topoisomerase 1 Activity to Favor Efficient Transcription. Cell 165: 357-371

Das BB, Huang SY, Murai J, Rehman I, Ame JC, Sengupta S, Das SK, Majumdar P, Zhang H, Biard D et al (2014) PARP1-TDP1 coupling for the repair of topoisomerase I-induced DNA damage. Nucleic Acids Res 42: 4435-4449

Das SK, Kuzin V, Cameron DP, Sanford S, Jha RK, Nie Z, Rosello MT, Holewinski R, Andresson T, Wisniewski J et al (2022) MYC assembles and stimulates topoisomerases 1 and 2 in a "topoisome". Mol Cell 82: 140-158 e112

Fabian Z, Kakulidis ES, Hendriks IA, Kuhbacher U, Larsen NB, Oliva-Santiago M, Wang J, Leng X, Dirac-Svejstrup AB, Svejstrup JQ et al (2024) PARP1-dependent DNA-protein crosslink repair. Nat Commun 15: 6641

Plo I, Liao ZY, Barcelo JM, Kohlhagen G, Caldecott KW, Weinfeld M, Pommier Y (2003) Association of XRCC1 and tyrosyl DNA phosphodiesterase (Tdp1) for the repair of topoisomerase I-mediated DNA lesions. DNA Repair (Amst) 2: 1087-1100

Sun Y, Chen J, Huang SN, Su YP, Wang W, Agama K, Saha S, Jenkins LM, Pascal JM, Pommier Y (2021) PARylation prevents the proteasomal degradation of topoisomerase I DNA-protein crosslinks and induces their deubiquitylation. Nat Commun 12: 5010

Reviewer #1:

The authors have addressed all my concerns.

Reviewer #2:

The authors have adequately revised the manuscript which is now suitable for publication to Nat Commun.

Reviewer #3:

The authors have adequately addressed all my comments and concerns by conducting several additional experiments and providing new data that support and extend their original conclusions. The revised manuscript is improved and is of significant interest to a broad scientific audience.

Response to the reviewers:

We would like to thank all reviewers for their time and valuable feedback. We are pleased to hear that we have addressed all of the reviewers' concerns, and that our study was considered to be of significant interest to a broad audience, suitable for publication in Nature Communications.